# Adaptive Denoising via GainTuning

**Sreyas Mohan**[1], **Joshua L. Vincent**[2], **Ramon Manzorro**[2], **Peter A. Crozier** [2],

**Carlos Fernandez-Granda**[1,3], **Eero P. Simoncelli**[1,3,4]

[1]Center For Data Science, NYU,
[2]School for Engineering of Matter, Transport and Energy, ASU
[3]Courant Institute of Mathematical Sciences, NYU
[4]Center for Neural Science, NYU and Flatiron Institute, Simons Foundation

## Abstract

Deep convolutional neural networks (CNNs) for image denoising are typically trained on large datasets. These models achieve the current state of the art, but they do not generalize well to data that deviate from the training distribution. Recent work has shown that it is possible to train denoisers on a single noisy image. These models adapt to the features of the test image, but their performance is limited by the small amount of information used to train them. Here we propose "GainTuning", a methodology by which CNN models pre-trained on large datasets can be adaptively and selectively adjusted for individual test images. To avoid overfitting, GainTuning optimizes a single multiplicative scaling parameter (the "Gain") of each channel in the convolutional layers of the CNN. We show that GainTuning improves state-of-the-art CNNs on standard image-denoising benchmarks, boosting their denoising performance on nearly every image in a held-out test set. These adaptive improvements are even more substantial for test images differing systematically from the training data, either in noise level or image type. We illustrate the potential of adaptive GainTuning in a scientific application to transmission-electron-microscope images, using a CNN that is pre-trained on synthetic data. In contrast to the existing methodology, GainTuning is able to faithfully reconstruct the structure of catalytic nanoparticles from these data at extremely low signal-to-noise ratios.

## 1 Introduction

Like many problems in image processing, the recovery of signals from noisy measurements has been revolutionized by the development of convolutional neural networks (CNNs) [66, 8, 67]. These models are typically trained on large databases of images, either in a supervised [37, 66, 8, 68, 67] or an unsupervised fashion [62, 3, 27, 29]. Once trained, these solutions are evaluated on noisy test images. This approach achieves state-of-the-art performance when the test images and the training data belong to the same distribution. However, when this is not the case, the performance of these models is often substantially degraded [59, 37, 68]. This is an important limitation for many practical applications, in which it is challenging (or even impossible) to gather a training dataset that is comparable in noise and signal content to the images encountered at test time. Overcoming this limitation requires *adaptation* to the test data.

A recent unsupervised method (Self2Self) has shown that CNNs can be trained exclusively on individual test images, producing impressive results [46]. Despite this, the performance of Self2Self is limited by the small amount of available training information, and is generally inferior to CNN models trained on large databases.

35th Conference on Neural Information Processing Systems (NeurIPS 2021).

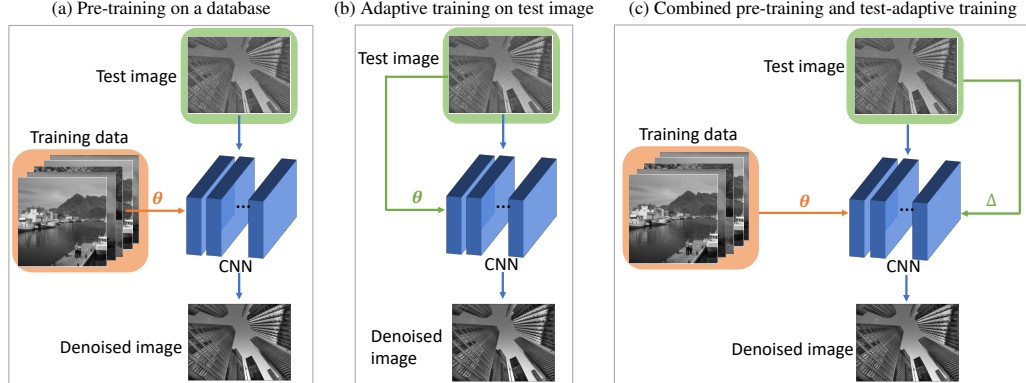

Figure 1: **Proposed denoising paradigm**. (a) Typically, CNNs are trained on a large dataset and evaluated directly on a test image. (b) Recent unsupervised methods perform training on a single test image. (c) We propose GainTuning, a framework which bridges the gap between both of these paradigms: a CNN pre-trained on a large training database is adapted to the test image.

In this work, we propose *GainTuning*, a framework to bridge the gap between models pre-trained on large datasets, and models trained exclusively on test images. In the spirit of two recent methods [55, 59], GainTuning adapts pre-trained CNN models to individual test images by minimizing an unsupervised denoising cost function, thus fusing the generic capabilities obtained from the training data with specific refinements matched to the structure of the test data. Rather than adapt the full parameter set (filter weights and additive constants) to the test image, GainTuning instead optimizes a single multiplicative scaling parameter (the "Gain") for each channel within each layer of the CNN. The dimensionality of this reduced parameter set is a small fraction ($\approx 0.1\%$ in our examples) of that of the full parameter set. We demonstrate through extensive examples that this prevents overfitting to the test data. The GainTuning procedure is general, and can be applied to any CNN denoising model, regardless of the architecture or pre-training process.

**Our contributions**. GainTuning provides a novel method for adapting CNN denoisers trained on large datasets to a single test image. GainTuning improves state-of-the-art CNNs on standard image-denoising benchmarks, boosting their denoising performance on nearly every image in held-out test sets. Performance improvements are even more substantial when the test images differ systematically from the training data. We showcase this ability through controlled experiments in which we vary the distribution of the noise and image structure of the test data. Finally, we evaluate GainTuning in a real scientific-imaging application where adaptivity is crucial: denoising transmission-electron-microscope data at extremely low signal-to-noise ratios. As shown in Figure 2, both CNNs pre-trained on simulated images and CNNs trained only on the test data produce denoised images with substantial artefacts. In contrast, GainTuning achieves effective denoising, accurately revealing the atomic structure in the real data.

## 2   Related Work

**Denoising via deep learning**. In the last five years, CNN-based methods have clearly outperformed previous state-of-the-art denoising methods [13, 53, 6, 45, 14, 21, 10]. Denoising CNNs are typically trained in a supervised fashion, minimizing mean squared error (MSE) over a large database of example ground-truth clean images and their noisy counterparts [66, 37, 8]. Unsupervised methods have also been developed, which do not rely on ground-truth images. There are two main strategies to achieve this: use of an empirical Bayes objective, such as Stein's unbiased risk estimator (SURE) [13, 33, 48, 36, 55, 56], and architectural "blind-spot" methods [27, 29, 3, 62] (see Section 4 for a more detailed description).

**Generalization to out-of-distribution noise**. Previous studies have shown that CNN denoisers fail to generalize when the noise encountered at test time differs from that of the training data [68, 37]. Ref. [37] proposes the use of a modified CNN architecture without additive bias terms, which is able to generalize to noise with variance well beyond that encountered in the training set. Here, we show

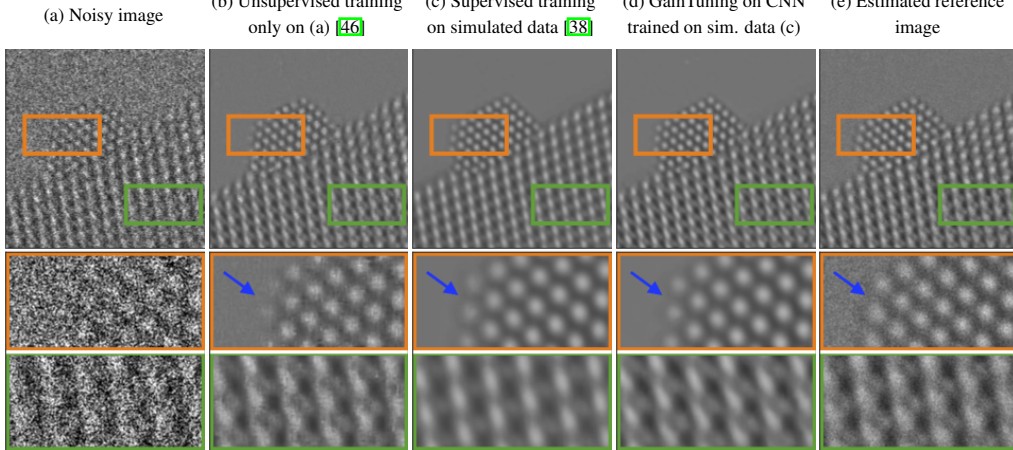

| (a) Noisy image | (b) Unsupervised training only on (a) [46] | (c) Supervised training on simulated data [38] | (d) GainTuning on CNN trained on sim. data (c) | (e) Estimated reference image |

Figure 2: **Denoising results for real-world data.** (a) An experimentally-acquired atomic-resolution transmission electron microscope image of a CeO2-supported Pt nanoparticle. The image has a very low signal to noise ratio (PSNR of $\approx 3dB$). (b) Denoised image obtained using Self2Self [46], which fails to reconstruct three atoms (blue arrow, second row). (c) Denoised image obtained via a CNN trained on a simulated dataset, where the pattern of the supporting atoms is not recovered faithfully (third row). (d) Denoised image obtained by adapting the CNN in (c) to the noisy test image in (a) using GainTuning. Both the nanoparticle and the support are recovered without artefacts. (e) Reference image, estimated by averaging $40$ different noisy images of the same nanoparticle.

that augmenting a generic architecture with GainTuning yields comparable performance to removing bias.

**Generalization to out-of-distribution images**. In order to adapt CNNs to operate on test data with characteristics differing from the training set, recent publications propose fine-tuning the networks using an additional training dataset that is more aligned with the test data [59, 18]. This is a form of transfer learning, a popular technique in classification problems [12, 64]. However, it is often challenging to obtain relevant additional training data. Here, we show that GainTuning can adapt CNN denoisers to novel test images.

**Feature normalization**. Normalization techniques such as batch normalization (BN) [23] are a standard component of deep CNNs. BN consists of two stages: (1) centering and normalizing the features corresponding to each channel, (2) scaling and shifting the normalized features using two learned parameters per channel (a scaling factor and a shift). The scaling parameter is analogous to the gain parameter introduced in GainTuning. However, in BN this parameter is adjusted during training and fixed during test time, whereas GainTuning adjusts it adaptively, for each test image.

**Gain normalization**. Motivated by gain control properties observed in biological sensory neurons [5], adaptive local normalization of response gains has been previously applied in object recognition [24], density estimation [1], and compression [2]. In contrast to these approaches, which adjust gains based on local responses, GainTuning adjusts a global gain for each channel by optimizing an unsupervised objective function.

**Adapting CNN denoisers to test data**. Two recent publications have developed methods of adapting CNN denoisers to test data [56, 59]. Ref. [56] include the noisy test images in the training set. In a recent extension, the authors fine-tune a pre-trained CNN on a single test image using the SURE cost function [55]. Ref. [59] does the same using a novel cost function based on noise resampling (see Section 4 for a detailed description). As shown in Section E fine-tuning the full set of CNN parameters using only a single test image can lead to overfitting. Ref. [55] avoids this using early stopping, selecting the number of fine-tuning steps beforehand. Ref. [59] uses a specialized architecture with a reduced number of parameters. Here, we show that several unsupervised cost functions can be used to perform adaptation without overfitting, as long as we only optimize a subset of the model parameters (specifically, the gain of each channel).

**Adjustment of channel parameters to improve generalization in other tasks**. Adjustment of channel parameters, such as gains and biases, has been shown to improve generalization in multiple machine-learning tasks, such as the vision-language problems [43, 11], image generation [7], style transfer [17], and image restoration [20]. In these methods, the adjustment is carried out while training the model by minimizing a supervised cost function. In image classification, recent studies have proposed performing adaptive normalization [25, 41, 51] and optimization [61] of channel parameters during test time, in the same spirit as GainTuning.

## 3 Proposed Methodology: GainTuning

In this section we describe the GainTuning framework. Let $f_\theta$ be a CNN denoiser parameterized by weight and bias parameters, $\theta$. We assume that we have available a training database and a test image $\mathbf{y}_{\text{test}}$ that we aim to denoise. First, the networks parameters are optimized on the training database

$$\theta_{\text{pre-trained}} = \arg\min_\theta \sum_{y \in \text{training database}} \mathcal{L}_{\text{pre-training}}(y, f_\theta(y)). \tag{1}$$

The cost function $\mathcal{L}_{\text{pre-training}}$ used for pre-training can be supervised, if the database contains clean and noisy examples, or unsupervised, if it only contains noisy data.

A direct method of adapting the pre-trained CNN to the test data is to finetune all the parameters, as is done in all prior work on test-time adaptation [59, 55, 18]. Unfortunately this can lead to *overfitting* the test data (see Section E). Due to the large number of degrees of freedom, the model is able to minimize the unsupervised cost function without denoising the noisy test data effectively. This can be avoided to some extent by employing CNN architectures with a small number of parameters [59], or by only optimizing for a short time ("early stopping") [55]. Unfortunately, using a CNN with reduced parameters can limit performance (see Section 5), and it is unclear how to choose a single criterion for early stopping that can operate correctly for all test images. Here, we propose a different strategy: tuning a single parameter (the *gain*) in each channel of the CNN. GainTuning can be applied to any pre-trained CNN.

We denote the gain parameter of the $c^{\text{th}}$ channel of the the $l^{\text{th}}$ layer as $\gamma[l, c]$, and the conventional parameters of that channel by $\theta_{\text{pre-trained}}[l, c]$ (a vector containing the filter weights). The adapted GainTuning parameters are the product of these:

$$\theta_{\text{GainTuning}}(\gamma)[l, c] = \gamma[l, c]\ \theta_{\text{pre-trained}}[l, c]. \tag{2}$$

We estimate the gains by minimizing an unsupervised loss that only depends on the noisy image:

$$\hat{\gamma} = \arg\min_\gamma\ \mathcal{L}_{\text{GainTuning}}(\mathbf{y}_{\text{test}}, \theta_{\text{GainTuning}}(\gamma)) \tag{3}$$

The final denoised image is $f_{\theta_{\text{GainTuning}}(\hat{\gamma})}(\mathbf{y}_{\text{test}})$. Section 4 describes several possible choices for the cost function $\mathcal{L}_{\text{GainTuning}}$. Since we use only one scalar parameter per channel, the adjustment performed by GainTuning is very low-dimensional ($\approx 0.1\%$ of the dimensionality of $\theta$). This makes optimization quite efficient, and prevents overfitting (see Section E). Further, in Section E we show that performing GainTuning provides better performance when compared to fine-tuning only the last few layers of the pre-trained network.

## 4 Cost Functions for GainTuning

A critical element of GainTuning is the use of an unsupervised cost function, which is minimized in order to adapt the pre-trained CNN to the test data. Here, we describe three different choices, each of which are effective for the GainTuning framework, but which have different benefits and limitations.

**Blind-spot loss**. This loss measures the ability of the denoiser to reproduce the noisy observation, while excluding the identity solution. To achieve this, the CNN must estimate the $j$th pixel $y_j$ of the noisy image $y$ as a function of the other pixels $y_{\{j\}^c}$, *excluding the pixel itself*. As long as the noise degrades pixels *independently*, the network to learn a nontrivial denoising function that exploits the relationships between pixels arising from the underlying clean image(s). The resulting loss can be written as

$$\mathcal{L}_{\text{blind-spot}}(\mathbf{y}, \theta) = \mathbb{E}\left[(f_\theta(\mathbf{y}_{\{j\}^c})_j - \mathbf{y}_j)^2\right]. \tag{4}$$

Here the expectation is over the data distribution and the selected pixel. This "blind spot" can be enforced through architecture design [29], or by masking [3, 27] (see also [46] and [62] for related approaches). The blind-spot loss has a key property that makes it very powerful in practical applications: it makes no assumption about the noise distribution beyond pixel-wise independence. When combined with GainTuning it achieves effective denoising of real electron-microscope data at very low SNRs (see Figure 2 and Section 5.4, F.5).

**Stein's Unbiased Risk Estimator (SURE).** Let $\mathbf{x}$ be an $N$-dimensional ground-truth random vector $\mathbf{x}$ and let $\mathbf{y} := \mathbf{x} + \mathbf{n}$ be a corresponding noisy observation, where $\mathbf{n} \sim \mathcal{N}(0, \sigma_n^2 \mathbf{I})$. SURE provides an expression for the MSE between $\mathbf{x}$ and a denoised estimate $f_\theta(\mathbf{y})$, which *only depends on the noisy observation* $\mathbf{y}$:

$$\mathbb{E}\left[\frac{1}{N}\|\mathbf{x} - f_\theta(\mathbf{y})\|^2\right] = \mathbb{E}\left[\frac{1}{N}\|\mathbf{y} - f_\theta(\mathbf{y})\|^2 - \sigma^2 + \frac{2\sigma^2}{N}\sum_{k=1}^{N}\frac{\partial(f_\theta(\mathbf{y})_k)}{\partial \mathbf{y}_k}\right] := \mathcal{L}_{\text{SURE}}(\mathbf{y}, \theta). \quad (5)$$

The last term in Equation 8 is the divergence of $f_\theta$, which can be approximated using Monte Carlo techniques [47] (Section D). The divergence is the sum of the partial derivatives of each denoised pixel with respect to the corresponding input pixel. Intuitively, penalizing it forces the denoiser to not rely as heavily on the $j$th noisy pixel to estimate the $j$th clean pixel. This is similar to the blind-spot strategy, with the added benefit that the $j$th noisy pixel is not ignored completely. To further illustrate this connection, consider a linear convolutional denoising function $f_\theta(\mathbf{y}) = \theta \circledast \mathbf{y}$, where the center-indexed parameter vector is $\theta = [\theta_{-k}, \theta_{-k+1}, \ldots, \theta_0, \ldots, \theta_{k-1}, \theta_k]$. The SURE cost function (Equation 8) reduces to

$$\mathbb{E}_{\mathbf{n}}\left[\frac{1}{N}\|\mathbf{y} - \theta \circledast \mathbf{y}\|^2\right] - \sigma^2 + 2\sigma^2\theta_0 \quad (6)$$

The SURE loss equals the MSE between the denoised output and the noisy image, with a penalty on the "self" pixel. As this penalty is increased, the self pixel will be ignored, so the loss tends towards the blind-spot cost function. When integrated into the GainTuning framework, the SURE loss is limited to additive Gaussian noise, for which it outperforms the blind-spot loss. Extensions of SURE to many other stochastic observation models have been developed [49], and may offer alternative objectives for GainTuning.

**Noise Resampling.** Ref. [59] introduced a novel procedure for adaptation which we call *noise resampling*. Given a pre-trained denoiser $f_\theta$ and a test image $\mathbf{y}$, first one obtains an initial denoised image by applying $f_\theta$ to $\mathbf{y}$, $\hat{\mathbf{x}} := f_{\theta_{\text{pre-trained}}}(\mathbf{y})$. This denoised image is then artificially corrupted $\hat{\mathbf{x}}$ by simulating noise from the same distribution as the data of interest to create synthetic noisy examples. Finally, the denoiser is fine-tuned by minimizing the MSE between $\hat{\mathbf{x}}$ and the synthetic examples. If we assume additive noise, the resulting loss is of the form

$$\mathcal{L}_{\text{noise resampling}}(\mathbf{y}, \theta) = \mathbb{E}_n\left[\|(f_\theta(\hat{\mathbf{x}} + \mathbf{n}) - \hat{\mathbf{x}}\|^2\right]. \quad (7)$$

Noise resampling is reminiscent of Refs. [40, 63], which add noise to an already noisy image. When integrated in the GainTuning framework, we find the noise-resampling loss results in effective denoising in the case of additive Gaussian noise, although it generally underperforms the SURE loss.

## 5 Experiments and Results

We performed three different types of experiment to evaluate the performance of GainTuning **In-distribution** (test examples held out from the training set); **out-of-distribution noise** (noise level or distribution of test examples differs from training set); and **out-of-distribution signal** (test images differ in features or context from the training set). We also apply GainTuning to **real data** from a transmission electron microscope.

Our experiments make use of four **datasets**: The BSD400 natural image database [34] with test sets Set12 and Set68 [66], the Urban100 images of urban environments [22], the IUPR dataset of scanned documents [4], and a set of synthetic piecewise constant images [31] (see Section B). We demonstrate the broad applicability of GainTuning by using it in conjunction with multiple **architectures for image denoising**: DnCNN [66], BFCNN [37], UNet [50] and Blind-spot net [29] (see Section A). Finally, we compare our results to several **benchmarks**: (1) models trained on the training database, (3) CNN models adapted by fine-tuning all parameters (as opposed to just the gains), (3) a model trained only on the test data, (4) LIDIA, a specialized architecture and adaptation strategy proposed in [59]. We provide details on training and optimization in Section C.

|  | Model | | Set12 | | | BSD68 | | |
|---|---|---|---|---|---|---|---|---|
|  |  |  | $\sigma = 30$ | 40 | 50 | 30 | 40 | 50 |
| GainTuning | DnCNN | Pre-trained | 29.52 | 28.21 | 27.19 | 28.39 | 27.16 | 26.27 |
|  |  | GainTuning | **29.62** | **28.30** | **27.29** | **28.47** | **27.23** | **26.33** |
|  | UNet | Pre-trained | 29.34 | 28.05 | 27.05 | 28.27 | 27.05 | 26.15 |
|  |  | GainTuning | 29.46 | 28.15 | 27.13 | 28.34 | 27.12 | 26.22 |
| Baseline | LIDIA | Pre-trained | 29.46 | 27.95 | 26.58 | 28.24 | 26.91 | 25.74 |
|  |  | Adapted | 29.50 | 28.10 | 26.95 | 28.23 | 26.97 | 26.02 |
|  | Self2Self |  | 29.21 | 27.80 | 26.58 | 27.83 | 26.67 | 25.73 |

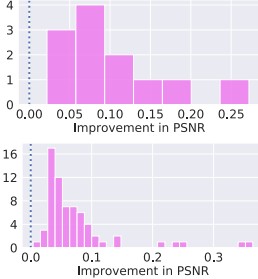

Figure 3: **GainTuning achieves state-of-the-art performance.** (Left) The average PSNR on two test set of generic natural images improves after GainTuning using SURE loss function for different architectures across multiple noise levels. The CNNs are trained on generic natural images (BSD400). (Right) Histograms of improvement in PSNR achieved by DnCNN over test images from Set12 (top) and BSD68 (bottom) at $\sigma = 30$.

## 5.1 GainTuning surpasses state-of-the-art performance for in-distribution data

**Experimental set-up**. We use BSD400, a standard natural-image benchmark, corrupted with Gaussian white noise with standard deviation $\sigma$ sampled uniformly from $[0, 55]$ (relative to pixel intensity range $[0, 255]$). Following [66], we evaluate performance on two independent test sets: Set12 and BSD68, corrupted with Gaussian noise with $\sigma \in \{30, 40, 50\}$.

**Comparison to pre-trained CNNs**. GainTuning consistently improves the performance of pre-trained CNN models. Figure 3 shows this for two different models, DnCNN [66] and UNet [50] (see also Section F.1). The SURE loss outperforms the blind-spot loss, and is slightly better than noise resampling (Table 7). The same holds for other architectures, as reported in Section F.1. On average the improvement is modest, but for some images it is quite substantial (up to 0.3 dB in PSNR for $\sigma = 30$, see histogram in Figure 3).

**Comparison to other baselines**. GainTuning outperforms fine-tuning based on optimizing all the parameters for different architectures and loss functions (see Section E). GainTuning clearly outperforms a Self2Self model, which is trained exclusively on the test data (Figure 3). It also outperforms the specialized architecture and adaptation process introduced in [59], with a larger gap in performance for higher noise levels.

## 5.2 GainTuning generalizes to new noise distributions

**Experimental set-up**. The same set-up as Section 5.1 is used, except that the test sets are corrupted with Gaussian noise with $\sigma \in \{70, 80\}$ (both beyond the training range of $\sigma \in [0, 55]$).

**Comparison to pre-trained CNNs**. Pre-trained CNN denoisers fail to generalize in this setting. GainTuning consistently improves their performance (see Figure 4).

The SURE loss again outperforms the blind-spot loss, and is slightly better than noise resampling (see Section F.2). The same holds for other architectures, as reported in Section F.2. The improvement in performance for all images is substantial (up to 12 dB in PSNR for $\sigma = 80$, see histogram in Figure 4).

**Comparison to other baselines**. GainTuning achieves comparable performance to a gold-standard CNN trained with supervision at all noise levels (Figure 4). GainTuning matches the performance of a bias-free CNN [37] specifically designed to generalize to out-of-distribution noise (Figure 4). GainTuning outperforms fine-tuning based on optimizing all the parameters for different architectures and loss functions (see Section E). GainTuning clearly outperforms a Self2Self model trained exclusively on the test data (Section F.2), and the LIDIA adaptation method [59].

**Gaussian to Poisson generalization**: Section F.2 and Figure 5 show that GainTuning can effectively adapt a CNN pre-trained for Gaussian noise removal to restore images corrupted with Poisson noise as well.

| | | **Out-of-distribution test noise** | | | |
|---|---|---|---|---|---|
| Test set | $\sigma$ | Trained on $\sigma \in [0, 55]$ | | Bias Free Model [37] | Trained on $\sigma \in [0, 100]$ |
| | | Pre-trained | Gaintuning | | |
| Set12 | 70 | 22.45 | 25.54 | 25.59 | 25.50 |
| | 80 | 18.48 | 24.57 | 24.94 | 24.88 |
| BSD68 | 70 | 22.15 | 24.89 | 24.87 | 24.88 |
| | 80 | 18.72 | 24.14 | 24.38 | 24.36 |

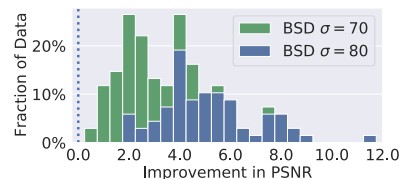

| | **Out-of-distribution test image** | | | |
|---|---|---|---|---|
| | Training data | Test data | Pre-trained | Gaintuning |
| (a) | Piecewise constant | Natural images | 27.31 | 28.60 |
| (b) | Natural images | Urban images | 28.35 | 28.79 |
| (c) | Natural images | Scanned documents | 30.02 | 30.73 |

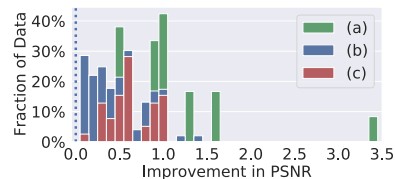

Figure 4: **GainTuning generalizes to out-of-distribution data.** Average performance of a CNN trained to denoise at noise levels $\sigma \in [0, 55]$ improves significantly on test image with noise outside the training range, $\sigma = 70, 80$ (top) and on images with different characteristics than training data (bottom) after GainTuning. Capability of GainTuning to generalize to out-of-distribution noise is comparable to that of Bias-Free CNN [37], which is an architecture explicitly designed to generalize to noise levels outside the training range, and to that of a denoiser trained with supervision at all noise levels. (Right) Histogram showing improvement in performance for each image in the test set. The improvement is substantial across most images, reaching nearly 12dB improvement in one example. For these examples, the denoiser was DnCNN (with additive bias terms) and the GainTuning loss function was SURE. See Section F.2 for experiments with other CNN architectures and loss functions.

## 5.3 GainTuning generalizes to out-of-distribution image content

**Experimental set-up**. We evaluate the performance of GainTuning on test images that have different characteristics from the training images. We perform the following controlled experiments:

(a) **Simulated piecewise constant images** → **Natural images**. We pre-train CNN denoisers on simulated piecewise constant images. These images consists of constant regions (of different intensities values) with the boundaries having varied shapes such as circle and lines with different orientations (see Section B for some examples). Piecewise constant images provide a crude model for natural images [35, 44, 31]. We use GainTuning to adapt a CNN trained on this dataset to generic natural images (Set12). This experiment demonstrates the ability of GainTuning to adapt from a simple simulated dataset to a significantly more complex real dataset.

(b) **Generic natural images** → **Images with high self-similarity**. We apply GainTuning to adapt a CNN trained on generic natural images to images in Urban100 dataset. Urban100 consists of images of buildings and other structures typically found in an urban setting, which contain substantially more repeating/periodic structure (see Section B) than generic natural images.

(c) **Generic natural images** → **Images of scanned documents**. We apply GainTuning to adapt a CNN trained on generic natural images to images of scanned documents in IUPR dataset (see Section B).

All CNNs were trained for denoising Gaussian white noise with standard deviation $\sigma \in [0, 55]$ and evaluated at $\sigma = 30$.

**Comparison to pre-trained CNNs**. GainTuning consistently improves the performance of pre-trained CNNs in all the three experiments. Figure 4 shows this for DnCNN when GainTuning is based on SURE loss. We obtain similar results with other architectures (see Section F.3). In experiment (a), all test images show substantial improvements over the pre-trained results (average increase of roughly 1.3dB, and best case more than 3 dB, at $\sigma = 30$). We observe similar trends for experiments (b) and (c) as well, with improvements being better on an average for experiment (c). Note that we obtain similar performance increases when both *image and noise are out-of-distribution* as discussed in Section F.4.

**Comparison to other baselines**. In experiment (a), GainTuning outperforms methods that optimize all parameters over different architectures and loss functions (Section E). However, Self2Self trained only on test data outperforms GainTuningin this case, because the test images contain content that differs substantially from the training images. Self2Self provides the strongest form of adaptation, since it is trained exclusively on the test image, whereas the denoising properties of GainTuning are partially due to the pretraining (see Sections 7, F.3). We did not evaluate LIDIA [59] for this experiment. For experiments (b) and (c), training all parameters clearly outperforms GainTuning for case (b), but has similar performance for (c). GainTuning outperforms LIDIA on experiments (b) and (c). Self2Self trained exclusively on test data outperforms GainTuning(and LIDIA) on (b) and (c) (see Sections 7, F.3).

## 5.4 Application to Electron microscopy

**Scientific motivation.** Transmission electron microscopy (TEM) is a popular imaging technique in materials science [54, 58]. Recent advancements in TEM enable to image at high frame rates [16, 15]. These images can for example capture the dynamic, atomic-level rearrangements of catalytic systems [57, 19, 30, 32, 9], which is critical to advance our understanding of functional materials. Acquiring image series at such high temporal resolution produces data severely degraded by shot noise. Consequently, there is an acute need for denoising in this domain.

**The need for adaptive denoising.** Ground-truth images are not available in TEM, because measuring at high SNR is often impossible. Prior work has addressed this by using simulated training data [38, 60], whereas others have trained CNNs directly on noisy real data [52].

**Dataset.** We use the training set of 5583 simulated images and the test set of 40 real TEM images from [38, 60]. The data correspond to a catalytic platinum nanoparticle on a $CeO_2$ support (Section B).

**Comparison to pre-trained CNN.** A CNN [29] pre-trained on the simulated data fails to reconstruct the pattern of atoms faithfully (green box in Figure 2 (c), (e)). GainTuning applied to this CNN using the blind-spot loss correctly recovers this pattern (green box in Figure 2 (d), (e)) reconstructing the small oxygen atoms in the $CeO_2$ support. GainTuning with noise resampling failed to reproduce the support pattern (probably because it is absent from the initial denoised estimate) (Section F.5).

**Comparison to other baselines.** GainTuning clearly outperforms Self2Self, which is trained exclusively on the real data. The denoised image from Self2Self shows missing atoms and substantial artefacts (see Section F.5). We also compare GainTuning dataset to blind-spot methods using the 40 test frames [29, 52]. GainTuning clearly outperforms these methods (see Section F.5). Finally, GainTuning outperforms fine-tuning based on optimizing all the parameters, which overfits heavily (see Section E).

# 6 Analysis

In this section, we perform a qualitative analysis of the properties of GainTuning.

**Which images benefit most from GainTuning adaptation?** Section G.1 shows the images in the different test datasets for which GainTuning achieves the most and the least improvement in PSNR. The result is quite consistent over multiple architectures: the improvement in performance achieved by GainTuning is larger if the test image contains highly repetitive patterns. This makes intuitive sense; the repetitions effectively provide multiple examples from which to learn these patterns during the unsupervised refinement.

**Generalization via GainTuning**. Section G.2 shows that GainTuning can achieve generalization to images that are similar to the test image used for adaptation.

**How does GainTuning adapt to out-of-distribution noise?** Generalization to out-of-distribution noise provides a unique opportunity to understand how GainTuning modifies the denoising function. Ref. [37] shows that the first-order Taylor approximation of denoising CNNs trained on multiple noise levels tend to have a negligible constant term, and that the growth of this term is the primary culprit for the failure of these models when tested on new noise levels. GainTuning reduces the amplitude of this constant term, facilitating generalization (See Section G.3 for more details).

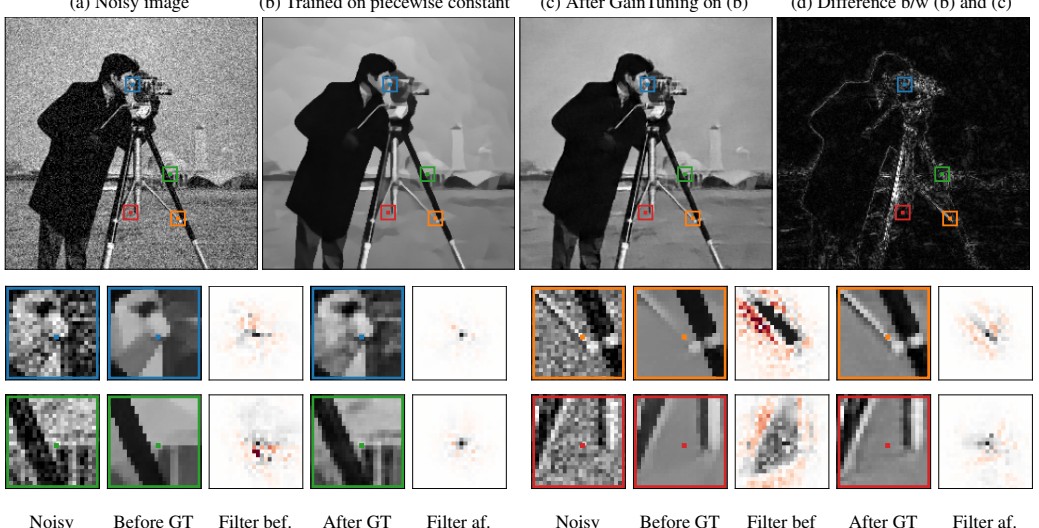

| (a) Noisy image | (b) Trained on piecewise constant | (c) After GainTuning on (b) | (d) Difference b/w (b) and (c) |

| Noisy | Before GT | Filter bef. | After GT | Filter af. | Noisy | Before GT | Filter bef | After GT | Filter af. |

Figure 5: **Adaptation to new image content**. (Top) A Bias-free CNN [37] pre-trained on piecewise constant images applied to a natural test image (a) oversmooths the image and blurs the details (b), but is able to recover more detail after applying GainTuning using SURE loss function (c). (Bottom) The CNN estimates a denoised pixel (colored pixel at the center of each image) as a linear combination of the noisy input pixels. The weighting functions (filters) of pre-trained CNN are dispersed, consistent with the training set. However, after GainTuning, the weighting functions are more precisely targeted to the local features, resulting in better recovery of details in the denoised image (c).

**How does GainTuning adapt to out-of-distribution images?** Figure 5 shows the result of applying a Bias-free CNN [37] trained on piecewise-constant images to natural images. Due to its learned prior, the CNN averages over large areas, ignoring fine textures. This is apparent in the equivalent linear filters obtained from a local linear approximation of the denoising function [37]. After GainTuning the model is better able to preserve the fine features, which is reflected in the equivalent filters (see Section G.4 for more details).

# 7   Limitations

As shown in Section 5, GainTuning improves the state of the art on benchmark datasets, adapts well to out-of-distribution noise and image content, and outperforms all existing methods on an application to real world electron-microscope data. A crucial component in the success of GainTuning is restricting the parameters that are optimized at test time. However, this constraint also limits the potential improvement in performance one can achieve, as seen when fine-tuning for test images from the Urban100 and IUPR datasets, each of which contain many images with highly repetitive structure. In these cases, we observe that fine-tuning all parameters, and even training only on the test data using Self2Self can outperform GainTuning. This raises the question of how to effectively leverage training datasets for such images.

In addition, when the pre-trained denoiser is highly optimized, and the test image is within distribution, GainTuning occasionally causes a slight degradation of performance. This is atypical (3 occurrences in 412 GainTuning experiments using DnCNN and SURE), and the decreases are quite small (maximum PSNR degradation of about 0.02dB, compared to maximum improvement of nearly 12dB; see Figure 14).

# 8   Conclusions

We've introduced GainTuning  an adaptive denoising methodology for adaptively fine-tuning a pre-trained CNN denoiser on individual test images. The method, which is general enough to be used with any denoising CNN, improves the performance of state-of-the-art CNNs on standard

denoising benchmarks, and provides even more substantial improvements when the test data differ systematically from the training data, either in noise level, noise type, or image type. We demonstrate the potential of adaptive denoising in scientific imaging through an application to electron microscopy. Here, GainTuning is able to jointly exploit synthetic data and test-time adaptation to reconstruct meaningful structure (the atomic configuration of a nanoparticle and its support), which cannot be recovered through alternative approaches. A concrete challenge for future research is to combine the unsupervised denoising strategy of Self2Self, which relies heavily on dropout and ensembling, with pre-trained models. More generally, it is of interest to explore whether GainTuning can provide benefits for other image-processing tasks.

Finally, we would like to comment on the potential negative societal outcomes of our work. The training of CNN models on large computational clusters contributes to carbon emissions, and therefore global warming. We hope that these effects may be offset to some extent by the potential applications of these approaches to tackle challenges such as global warming. In particular, the catalytic system studied in this work is representative of catalysts used in clean energy conversion and environmental remediation [39, 65, 42].

## Acknowledgments and Disclosure of Funding

We gratefully acknowledge financial support from the National Science Foundation (NSF): NSF NRT HDR Award 1922658 partially supported SM. NSF CBET 1604971 supported JLV and PAC, and NSF OAC-1940263 supported RM and PAC. NSF OAC-1940097 and OAC-2103936 supported CFG. Funding from the Simons Foundation supported SM and EPS. Thanks to ASU Research Computing and NYU HPC for high performance computing resources, and the John M. Cowley Center for High Resolution Electron Microscopy at Arizona State University.

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
