# A CNN architectures

In this section we describe the denoising architectures used for our computational experiments. All architectures except BFCNN have additive (bias) terms after every convolutional layer.

## A.1 DnCNN

DnCNN [66] consists of 20 convolutional layers, each consisting of $3 \times 3$ filters and 64 channels, batch normalization [23], and a ReLU nonlinearity. It has a skip connection from the initial layer to the final layer, which has no nonlinear units.

## A.2 BFCNN

We use BFCNN [37] based on DnCNN architecture, i.e, we remove all sources of additive bias, including the mean parameter of the batch-normalization in every layer (note however that the scaling parameter is preserved).

## A.3 UNet

Our UNet model [50] has the following layers:

1. *conv1* - Takes in input image and maps to 32 channels with $5 \times 5$ convolutional kernels.
2. *conv2* - Input: 32 channels. Output: 32 channels. $3 \times 3$ convolutional kernels.
3. *conv3* - Input: 32 channels. Output: 64 channels. $3 \times 3$ convolutional kernels with stride 2.
4. *conv4*- Input: 64 channels. Output: 64 channels. $3 \times 3$ convolutional kernels.
5. *conv5*- Input: 64 channels. Output: 64 channels. $3 \times 3$ convolutional kernels with dilation factor of 2.
6. *conv6*- Input: 64 channels. Output: 64 channels. $3 \times 3$ convolutional kernels with dilation factor of 4.
7. *conv7*- Transpose Convolution layer. Input: 64 channels. Output: 64 channels. $4 \times 4$ filters with stride 2.
8. *conv8*- Input: 96 channels. Output: 64 channels. $3 \times 3$ convolutional kernels. The input to this layer is the concatenation of the outputs of layer *conv7* and *conv2*.
9. *conv9*- Input: 32 channels. Output: 1 channels. $5 \times 5$ convolutional kernels.

The structure is the same as in [68]. This configuration of UNet assumes even width and height, so we remove one row or column from images in with odd height or width.

## A.4 Blind-spot network

We use a modified version of the blind-spot network architecture introduced in Ref. [29]. We rotate the input frames by multiples of $90°$ and process them through four separate branches (with shared weights) containing asymmetric convolutional filters that are *vertically causal*. The architecture of a branch is described in Table 1. Each branch has one input channel and one output channel. Each branch is followed by a de-rotation and the output is passed to a series of three cascaded $1 \times 1$ convolutions and non-linearity for reconstruction with 4 and 96 intermediate output channels, as in [29]. The final convolutional layer is linear and has 1 output channel.

# B Datasets

We perform controlled experiments on datasets with different signal and noise structure to evaluate the broad applicability of GainTuning (see Figure 6 for a visual summary of datasets). We describe each dataset below:

**Generic natural images.** We use 400 images from BSD400 [34] dataset for pre-training CNNs. We evaluate on two test sets, Set12 and Set68, with 12 and 68 images, respectively [66].

**Images of urban scenes.** We evaluate generalization capabilities of GainTuning using a dataset of images captured in urban settings, Urban100 [22]. These images often contain repeating patterns and structures, unlike generic natural images (see Figure 6). We evaluate GainTuning on the first 50 images from this dataset.

**Images of scanned documents.** We use images of scanned documents from the IUPR dataset [4]. We resized the images in IUPR dataset by a factor of 6, and used the first 50 images from the dataset for evaluation.

**Simulated piecewise constant images.** We use a dataset of simulated piecewise constant images. These images have constant regions with boundaries consisting of various shapes such as circles and lines with different

| Name | $N_{out}$ | Function |
|---|---|---|
| Input | 1 | |
| enc_conv_0 | 48 | Convolution $3 \times 3$ |
| enc_conv_1 | 48 | Convolution $3 \times 3$ |
| enc_conv_2 | 48 | Convolution $3 \times 3$ |
| pool_1 | 48 | MaxPool $2 \times 2$ |
| enc_conv_3 | 48 | Convolution $3 \times 3$ |
| enc_conv_4 | 48 | Convolution $3 \times 3$ |
| enc_conv_5 | 48 | Convolution $3 \times 3$ |
| pool_2 | 48 | MaxPool $2 \times 2$ |
| enc_conv_6 | 96 | Convolution $3 \times 3$ |
| enc_conv_7 | 96 | Convolution $3 \times 3$ |
| enc_conv_8 | 48 | Convolution $3 \times 3$ |
| upsample_1 | 48 | NearestUpsample $2 \times 2$ |
| concat_1 | 96 | Concatenate output of pool_1 |
| dec_conv_0 | 96 | Convolution $3 \times 3$ |
| dec_conv_1 | 96 | Convolution $3 \times 3$ |
| dec_conv_2 | 96 | Convolution $3 \times 3$ |
| dec_conv_3 | 96 | Convolution $3 \times 3$ |
| upsample_2 | 96 | NearestUpsample $2 \times 2$ |
| concat_2 | $96+k_1$ | Concatenate output of Input |
| dec_conv_4 | 96 | Convolution $3 \times 3$ |
| dec_conv_5 | 96 | Convolution $3 \times 3$ |
| dec_conv_6 | 96 | Convolution $3 \times 3$ |
| dec_conv_7 | 1 | Convolution $3 \times 3$ |

Table 1: **Blind-spot network**. The convolution and pooling layers are the blind-spot variants described in Ref. [29].

orientations. The constant region has an intensity value sampled from a uniform distribution between 0 and 1 (see Figure 6). These piecewise constant images provide a crude model for natural images [35, 44, 31], and a CNN pre-trained on this dataset provides a substrate for testing the ability of GainTuning to adapt to the complexity of real-world images.

**Simulated transmission electron microscopy data**. The TEM image data used in this work correspond to images from a widely utilized catalytic system, which consist of platinum (Pt) nanoparticles supported on a larger cerium (IV) oxide ($CeO_2$) nanoparticle. We use the simulated TEM image dataset introduced in Ref. [38] for pre-training CNNs. The simulated dataset contains 1024 x 1024 images, which are binned to match the approximate pixel size of the experimentally acquired real image series (described below). To equate the intensity range of the simulated images with those acquired experimentally, the intensities of the simulated images were scaled by a factor which equalized the vacuum intensity in a single simulation to the average intensity measured over a large area of the vacuum in a single 0.025 second experimental frame (i.e., 0.45 counts per pixel in the vacuum region). Furthermore, during TEM imaging multiple electron-optical and specimen parameters can give rise to complex, non-linear modulations of the image contrast. These parameters include the objective lens defocus, the specimen thickness, the orientation of the specimen, and its crystallographic shape/structure. Various combinations of these parameters may cause the contrast of atomic columns in the image to appear as black, white, or an intermediate mixture of the two. To account for this, the simulated dataset contains various instances of defocus, tilt, thickness, and shape/structure. We refer interested readers to Ref. [38] for more details.

**Real transmission electron microscopy data**. The real data consist of a series of images of the $Pt/CeO_2$ catalyst. The images were acquired in a $N_2$ gas atmosphere using an aberration-corrected FEI Titan transmission electron microscope (TEM), operated at 300 kV and coupled with a Gatan K2 IS direct electron detector [38]. The detector was operated in electron counting mode with a time resolution of 0.025 sec/frame and an incident electron dose rate of 5,000 $e^-/\text{Å}^2$/s. The electromagnetic lens system of the microscope was tuned to achieve a highly coherent parallel beam configuration with minimal low-order aberrations (e.g., astigmatism, coma), and a third-order spherical aberration coefficient of approximately -13 $\mu$m. We refer interested readers to Ref. [38] for more details.

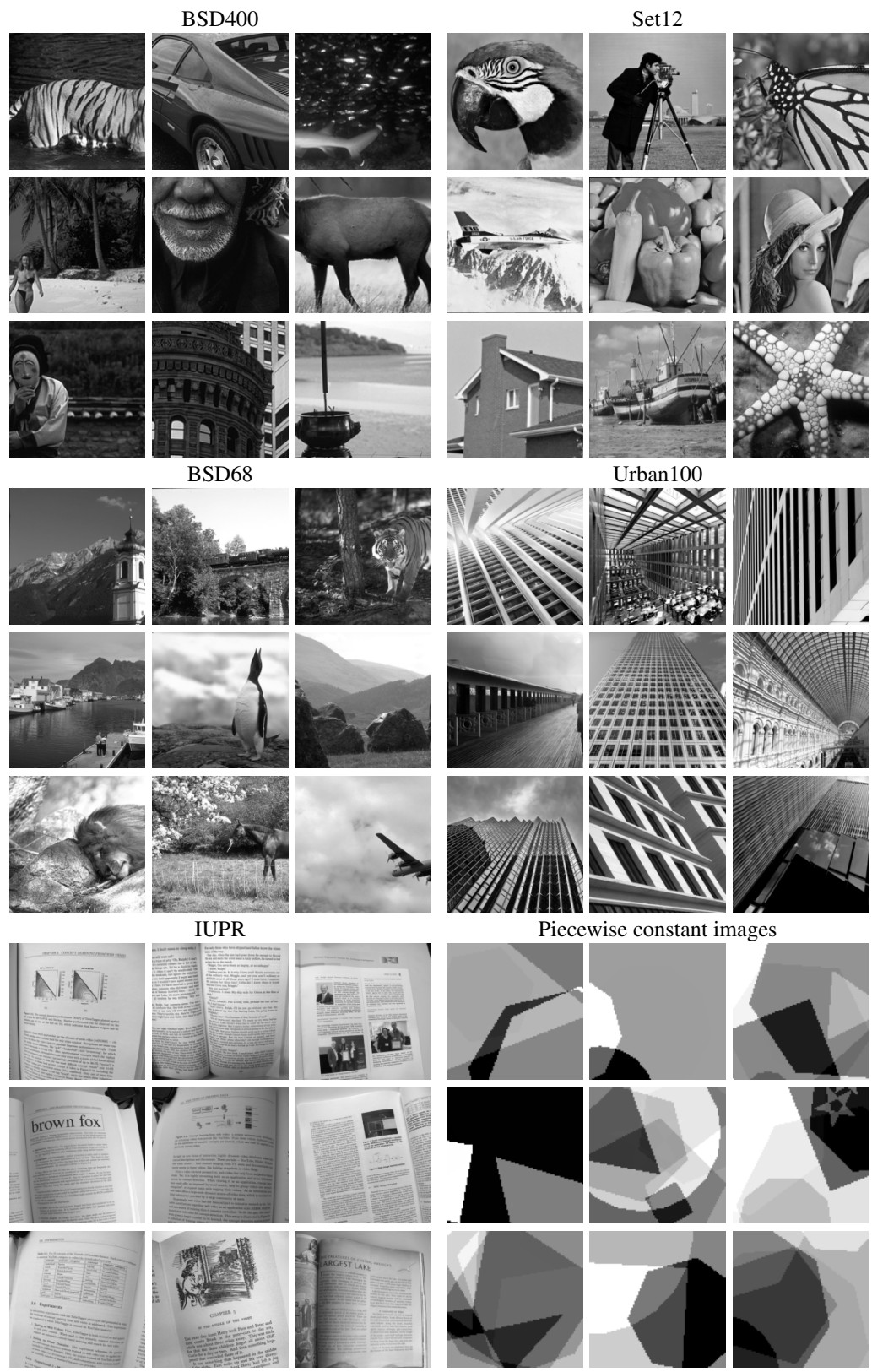

Figure 6: **Example dataset images**. Nine images chosen at random from each dataset.

# C  Details of pre-training and GainTuning

In this section, we describe the implementation details of the pre-training process and our proposed GainTuning framework.

## C.1  Overview

While performing GainTuning, we introduce a scalar multiplicative parameter (gain) in every channel of the convolutional layers in the denoising CNN. We do not introduce gain parameters in the last layer of the network. We describe the general optimization process for GainTuning here, and describe any additional modifications for specific datasets in the respective subsections.

**Data**. We perform GainTuning on patches extracted from the noisy image. We extracted $400 \times 400$ patches for the electron microscopy dataset, and $50 \times 50$ patches for all other datasets. We do not perform any data augmentation on the extracted patches.

**BatchNorm layers during GainTuning**. If the denoising CNN contains batch normalization (BN) layers (only DnCNN [66] and BFCNN [37] in our experiments), we freeze their statistics while performing GainTuning. That is, we do not re-estimate the mean and standard deviation parameter for each layer from the test data. Instead, we re-use the original values estimated from pre-training dataset.

**Optimization for GainTuning**. We use Adam [26] optimizer. We empirically find that training for 100 steps with a starting learning rate of $10^{-4}$ which is reduced to $10^{-5}$ after the $20^{\text{th}}$ step performs well across most situations (see sections below for hyper-parameters used in different experiments). Here, we define each step as a pass through 5000 random patches extracted from the test image. When performing experiments which compare optimizing all parameters to optimizing only gain during the adaptation process, we kept the learning rate constant at $10^{-5}$ for both options, and trained for 1000 steps.

## C.2  Natural images

**Pre-training dataset**. Our experiments are carried out on $180 \times 180$ natural images from the Berkeley Segmentation Dataset [34]. We use a training set of $400$ images. The training set is augmented via downsampling, random flips, and random rotations of patches in these images [66]. We train the CNNs on patches of size $50 \times 50$, which yields a total of 541,600 clean training patches.

**Pre-training process**. We train DnCNN, BFCNN and UNet using the Adam Optimizer [26] for 100 epochs with an initial learning rate of $10^{-3}$ and a decay factor of 0.5 for every 10 epochs after the $50^{th}$, with no early stopping [37].

**GainTuning**. We follow the same procedure as Section C.1.

## C.3  Piecewise constant images

**Pre-training dataset**. We generated a synthetic dataset of piecewise constant images with the varied boundary shapes like slanted lines and circles (see Figure 6). The intensity values of the constant regions were uniformly sampled between 0 and 1. The generated patches were of size $50 \times 50$ to mimic the training process for natural images [66].

**Pre-training**. We train DnCNN, BFCNN and UNet using the Adam Optimizer [26] using the same process as in Section C.2. For each epoch, we generated $50,000$ random patches.

**GainTuning** . We follow the same procedure as Section C.1.

## C.4  Electron microscope data

**Pre-training dataset**. Our experiments are carried out on $400 \times 400$ patches extracted from about 5000 simulated TEM introduced in Ref. [38]. The training set is augmented via downsampling, random flips, and random rotations of patches in these images [38, 60].

**Optimization Details:** We trained using Adam [26] optimizer with a starting learning of $10^{-4}$. The learning rate was decreased by a factor of 2 at checkpoints $[20, 25, 30]$ during a total training of 40 epochs [38].

**GainTuning**. We performed GainTuning using Adam [26] optimizer with a constant learning rate of $10^{-5}$ for 100 steps. Each step consisted of 1000 randomly sampled patches of size $400 \times 400$ extracted from the test image.

## C.5 Computational resources used

The computations were performed on an internal cluster equipped with NVIDIA RTX8000 and NVIDIA V100 GPUs. We used open-source pre-trained networks when available.

## D  Approximation for SURE

Let $\mathbf{x}$ be an $N$-dimensional ground-truth random vector $\mathbf{x}$ and let $\mathbf{y} := \mathbf{x} + \mathbf{n}$ be a corresponding noisy observation, where $\mathbf{n} \sim \mathcal{N}(0, \sigma_n^2 \mathbf{I})$. Stein's Unbiased Risk Estimator (SURE) provides an expression for the mean-squared error between $\mathbf{x}$ and the denoised estimate $f_\theta(\mathbf{y})$ (where $f_\theta$ denotes an arbitrary denoising function), which *only depends on the distribution of noisy observations* $\mathbf{y}$:

$$\mathbb{E}_{\mathbf{x},\mathbf{y}} \left[ \frac{1}{N} \|\mathbf{x} - f_\theta(\mathbf{y})\|^2 \right] = \mathbb{E}_{\mathbf{y}} \left[ \frac{1}{N} \|\mathbf{y} - f_\theta(\mathbf{y})\|^2 - \sigma^2 + \frac{2\sigma^2}{N} \sum_{k=1}^{N} \frac{\partial(f_\theta(\mathbf{y})_k)}{\partial \mathbf{y}_k} \right] \tag{8}$$

The last (divergence) term in the equation is costly to compute. Therefore, we make use of a Monte Carlo approximation of SURE introduced by Ref. [47]:

$$\sum_{k=1}^{N} \frac{\partial(f_\theta(\mathbf{y})_k)}{\partial \mathbf{y}_k} \approx \frac{1}{\epsilon N} \langle \tilde{\mathbf{n}}, f_\theta(\mathbf{y} + \epsilon \tilde{\mathbf{n}}) - f_\theta(\mathbf{y}) \rangle \tag{9}$$

where $\langle \mathbf{x}, \mathbf{y} \rangle$ represents the dot product between $\mathbf{x}$ and $\mathbf{y}$, $\tilde{n}$ represents a sample from $\mathcal{N}(0, 1)$, and $\epsilon$ represents a fixed, small, positive number. We set $\epsilon = \sigma \times 1.4 \times 10^{-4}$ for our computational experiments [55]. Equation (9) has been used in the implementation of several traditional [47], and deep learning based [36, 56, 55] denoisers.

## E  GainTuning prevents overfitting

We perform controlled experiments to compare test-time updating of (1) all parameters of a CNN, and (2) only the gain parameters. We briefly describe each experiment and our findings in this section.

**Comparison across different cost functions**. We fine-tune (a) all parameters, and (b) only gain parameters of a DnCNN [66] model when the test image is (1) in-distribution, (2) corrupted with out-of-distribution noise and (c) contains image features which are different from the training set. Fine-tuning only the gain parameters outperforms fine-tuning all parameters in all of these situations for different choices of cost functions (see Figures 7, 8 and 9)

**Comparison across different architectures**. We fine-tune (a) all parameters, and (b) only gain parameters of a DnCNN [66], BFCNN [37] and, UNet [50] model when the test image is (a) in-distribution, (b) corrupted with out-of-distribution noise and (c) contains image features which are different from the training set. Fine-tuning only the gain parameters often outperforms fine-tuning all parameters in all of these situations for different choices of cost functions (see Figure 10). Figure 10 shows results for a CNN trained on generic natural images and tested on images of urban scenes. In this case, training all parameters of the CNN outperforms training only the gains (see Section 7 for a discussion). Interestingly, training gains is comparable to training all parameters when we corrupt the images from urban scenes with a noise level that is also outside the training range (see Figure 11).

**GainTuning does not require early stopping**. Optimizing all parameters of a CNN during adaptation often results in overfitting (see Figure 10). In contrast, optimizing only the gain parameters for longer periods of time results improves performance without overfitting (Figure 12).

**Real electron microscopy data**. We fine-tune (a) all parameters, and (b) gain parameters to adapt a CNN to real images of nanoparticle acquired through an electron microscope. The CNN was pre-trained on the simulated data described in Section B. Optimizing only the gain parameters outperforms optimizing all parameter and does not require early stopping (Figure 13)

**GainTuning outperforms fine-tuning last few layers of the CNN**. We compared GainTuning to selectively fine-tuning last $n$ layers for DnCNN with $n = 20$ layers. GainTuning out-performed fine-tuning last layers by a substantial margin (see Table 2 for details). Note that gains only constitute 1.15K or 0.17% of the parameters, while fine-tuning only the last 2 layers is 37K or 5.63% parameters (about 33x more than the number of gains). The in-distribution and out-of-distribution noise consists of adapting a DnCNN trained on natural images with $\sigma \in [0, 55]$ for natural images (Set12) with $\sigma = 30$ and $\sigma = 70$ respectively. We adapted a CNN trained on piecewise constant images with $\sigma \in [0, 55]$ to natural images (Set12) with $\sigma = 30$ for out-of-distribution signal experiments.

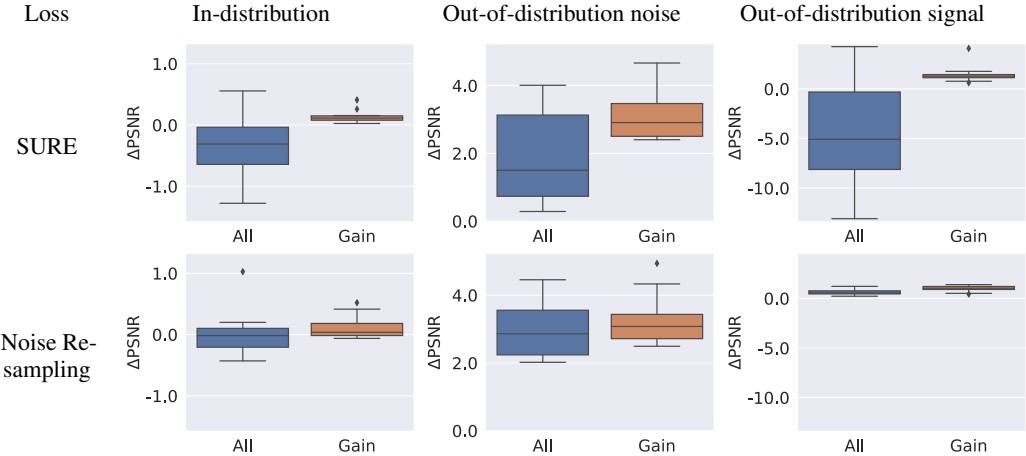

Figure 7: **GainTuning prevents overfitting.** Comparison of adaptive training of all network parameters, and GainTuning (training of gains only), using two different unsupervised objectives - SURE (top) and noise resampling (bottom). The distributions of performance improvements are shown as box plots. See Figure 8 for corresponding scatterplots. For *in-distribution*, we evaluate a CNN pre-trained on natural images corrupted with Gaussian noise of standard deviation $\sigma \in [0, 55]$ on natural images (Set12) at $\sigma = 30$. For *out-of-distribution noise* we evaluate natural images (Set12) at $\sigma = 70$. For *out-of-distribution signal* we evaluate a CNN trained on piecewise constant images at $\sigma \in [0, 55]$ on natural images (set12) at $\sigma = 30$. Please refer to Section F for details.

| | | Fine-tuning | | | | | |
| | All params | Last $n$ layers | | | | | Only gains |
| | | $n = 10$ | $n = 4$ | $n = 3$ | $n = 2$ | $n = 1$ | |
|---|---|---|---|---|---|---|---|
| Num. params (% of total params) | 668,225 (100%) | 334,081 (49.95%) | 111,745 (16.72%) | 74,689 (11.18%) | 37,633 (5.63%) | 577 (0.09%) | 1,152 (0.17%) |
| in-distribution | -0.33 | 0.09 | 0.05 | 0.04 | 0.04 | 0.06 | **0.14** |
| out-of-distr. noise | 1.92 | 1.92 | 2.05 | 2.06 | 2.10 | 2.13 | **3.11** |
| out-of-distr. signal | -4.48 | 0.92 | 1.12 | 1.06 | 0.93 | 0.83 | **1.45** |

Table 2: **GainTuning vs selectively fine-tuning last few layers.** We compared GainTuning to selectively fine-tuning last $n$ layers for a DnCNN with $n = 20$ layers. Table entries indicate the change in performance (i.e., the performance in PSNR after fine-tuning minus the PSNR of the pre-trained network - larger positive values are better). Across different tasks, GainTuning outperformed fine-tuning last layers by a significant margin. The in-indistribution and out-of-distribution signal consists of adapting a DnCNN trained on natural images with $\sigma \in [0, 55]$ for natural images (Set12) with $\sigma = 30$ and $\sigma = 70$ respectively. We adapted a CNN trained on piecewise constant images with $\sigma \in [0, 55]$ to natural images (Set12) with $\sigma = 30$ for out-of-distribution signal experiments.

## F  Performance of GainTuning

### F.1  In-distribution test image

**Different architectures.** We evaluated DnCNN [66], UNet [50] and BFCNN [37] architectures for this task. All models were trained on denoising Gaussian white noise of standard deviation $\sigma \in [0, 55]$ from generic natural images. Results of DnCNN and UNet are presented in Figure 3 in the main paper. Results for the BFCNN architecture are provided in Table 3.

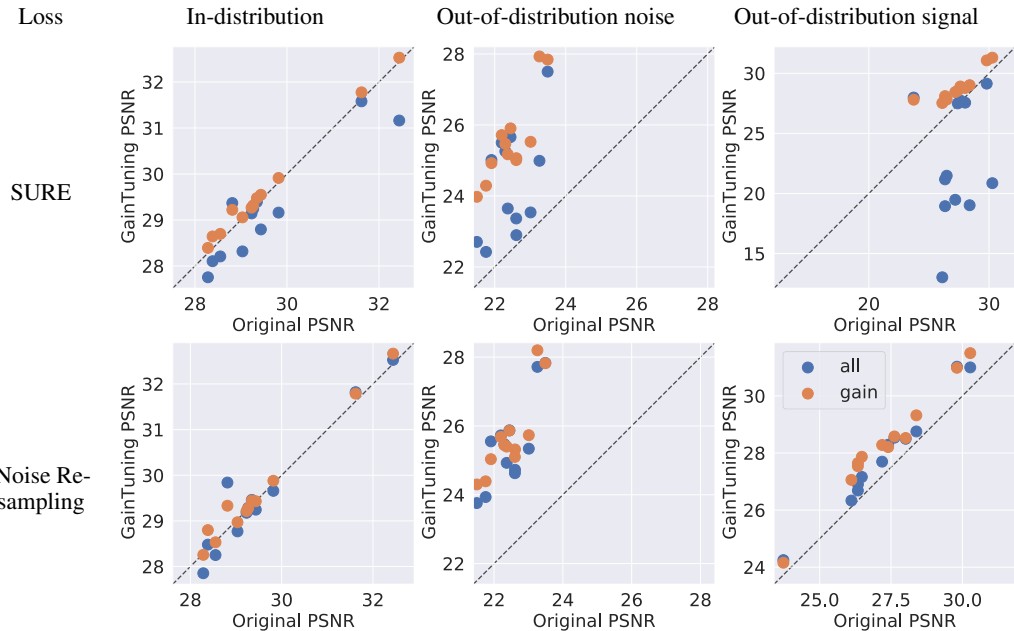

Figure 8: **GainTuning prevents overfitting.** Performance obtained from adaptively training all network parameters (blue points), compared to GainTuning (orange points) using the SURE loss, plotted against performance of the originally trained network. Each data point corresponds to one image in the dataset. The dashed line represents the identity (i.e., no improvement). Training all parameters (blue points) often leads to degraded performance, but training only the gains (orange points), leads to an improvement. For *in-distribution* test images, we evaluate a CNN pre-trained on natural images corrupted with Gaussian noise of standard deviation $\sigma \in [0, 55]$ on natural images (Set12) at $\sigma = 30$. For *out-of-distribution noise* we test on natural images (Set12) at $\sigma = 70$. For *out-of-distribution signal* we test a CNN trained on piecewise constant images at $\sigma \in [0, 55]$ on natural images (set12) at $\sigma = 30$. Please refer to Section F for details.

| Model | $\sigma$ | Set12 | | Set68 | |
|---|---|---|---|---|---|
| | | Pre-trained | GainTuning | Pre-trained | GainTuning |
| BFCNN | 30 | 29.52 | 29.61 | 28.36 | 28.45 |

Table 3: **Results for BFCNN**. Results for BFCNN [37] architecture trained on BSD400 dataset corrupted with Gaussian noise of standard deviation $\sigma \in [0, 55]$. Results for other architectures are provided in Section 5.1.

**Different cost functions**. We provide the results of evaluating DnCNN architecture with different cost functions in Table 7.

**Distribution of improvements.** We visualize the distribution of improvements in denoising performance for different architectures after performing GainTuning using the SURE cost function in Figure 14. As discussed in Section 7, if the CNN is optimized well and the test image is in-distribution, GainTuning can degrade performance. This degradation is atypical (3 out of 408 total evaluations) and very small (maximum degradation of 0.02 dB in PSNR).

## F.2 Out-of-distribution noise

**Different Architectures**. We summarize the results using DnCNN in Table 4 in the main paper. Figure 10 shows that the UNet architecture is also able to generalize to out-of-distribution noise.

**Different Loss Functions**. We provide the results of evaluating DnCNN architecture with different cost functions in Table 7.

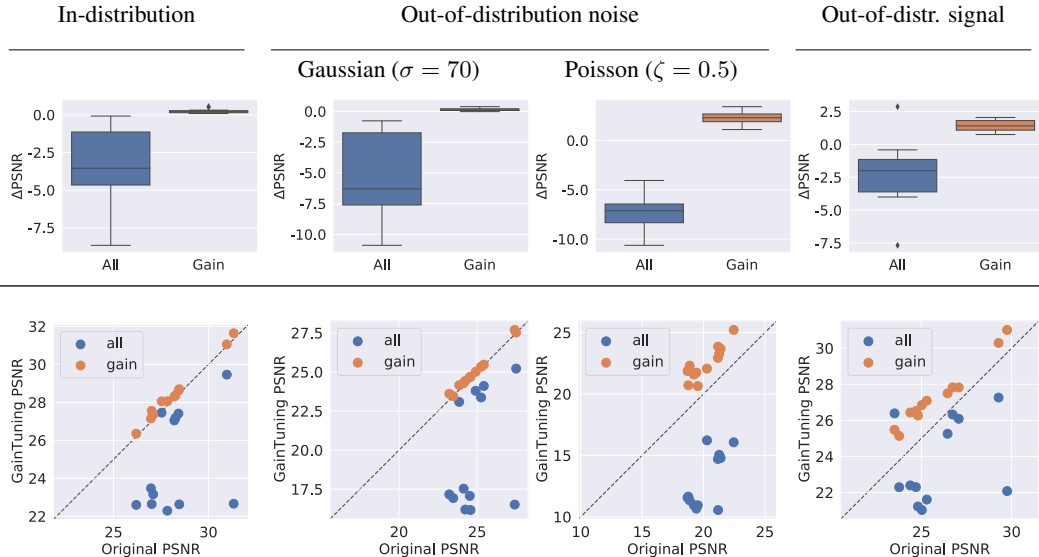

Figure 9: **GainTuning prevents overfitting.** Comparison of adaptive training of all network parameters, and GainTuning (training of gains only) using **blind-spot** cost function. The distribution of the gain in performance is visualized as a box plot. For *in-distribution*, we evaluate a CNN pre-trained on natural images corrupted with Gaussian noise of standard deviation $\sigma \in [0, 55]$ on natural images (Set12) at $\sigma = 30$. For *out-of-distribution noise* we evaluate natural images (Set12) at $\sigma = 70$ (Gaussian noise), and $\zeta = 0.5$ for Poisson noise. For *out-of-distribution signal* we evaluate a CNN trained on piecewise constant images at $\sigma \in [0, 55]$ on natural images (Set 12) at $\sigma = 30$. We used network architecture in [29] for our experiments.

| Test set | $\sigma$ | Trained on $\sigma \in [0, 55]$ | | Baselines | | | | |
| | | | | Bias Free Model [37] | Trained on $\sigma \in [0, 100]$ | LIDIA [59] | | S2S [46] |
| | | Pre-trained | GainTuning | | | Pre-trained | Adapted | |
| Set12 | 70 | 22.45 | 25.54 | 25.59 | 25.50 | 23.69 | 25.01 | 24.61 |
| | 80 | 18.48 | 24.57 | 24.94 | 24.88 | 22.12 | 24.17 | 23.64 |
| BSD68 | 70 | 22.15 | 24.89 | 24.87 | 24.88 | 23.28 | 24.57 | 24.29 |
| | 80 | 18.72 | 24.14 | 24.38 | 24.36 | 21.87 | 23.97 | 23.65 |

Table 4: **GainTuning for out-of-distribution noise**. We evaluate a DnCNN trained on generic natural images for $\sigma \in [0, 55]$ on a test set of generic natural images corrupted with $\sigma = \{70, 80\}$, which is outside the training range of the network. GainTuning is able is generalize effectively to this out-of-distribution test set. GainTuning achieves comparable performance to a network trained with supervision on a large range of noise levels ($\sigma \in [0, 100]$) an bias-free models which is an architecture explicitly designed to generalize to noise levels outside the training range. GainTuning also outperforms LIDIA [59], a specialized architecture and adaptation procedure, and Self2Self [46], a method trained exclusively on the test image.

**Comparison to baselines**. Table 4 summarizes the result of evaluating a DnCNN trained on generic natural images for $\sigma \in [0, 55]$ on a test set of generic natural images corrupted with $\sigma = \{70, 80\}$, which is outside the training range of the network. GainTuning is able to generalize effectively to this out-of-distribution test set. GainTuning achieves comparable performance to a network trained with supervision on a large range of noise levels ($\sigma \in [0, 100]$), and a bias-free model which is explicitly designed to generalize to noise levels outside the training range. GainTuning also outperforms LIDIA [59] (a specialized architecture and adaptation procedure). and Self2Self [46] (a method trained exclusively on the test image).

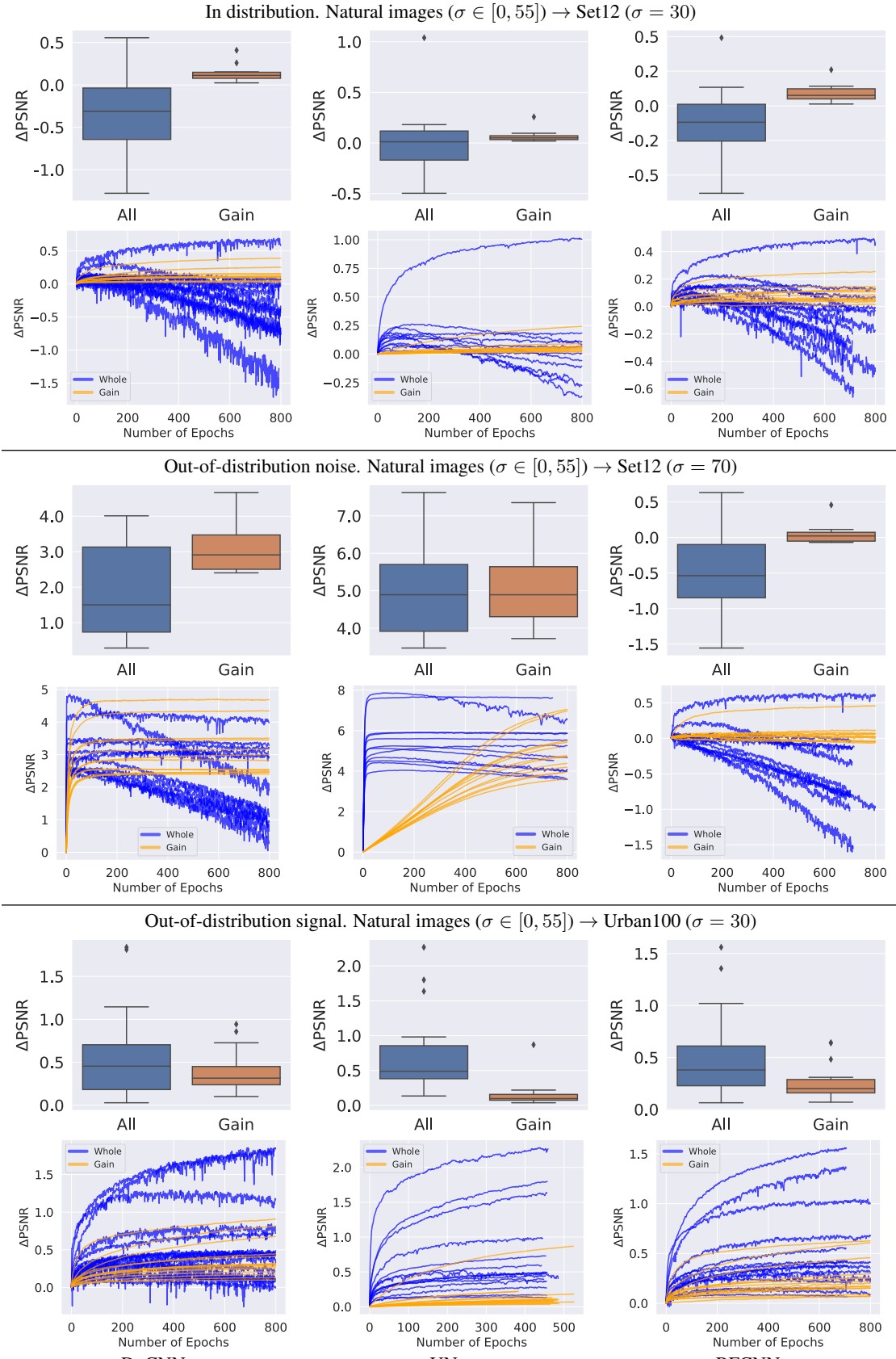

Figure 10: **GainTuning prevents overfitting**. We compare training all parameters of the network (blue) and only the gain parameters (orange) during the adaptation process. All architectures are trained using the SURE cost function.

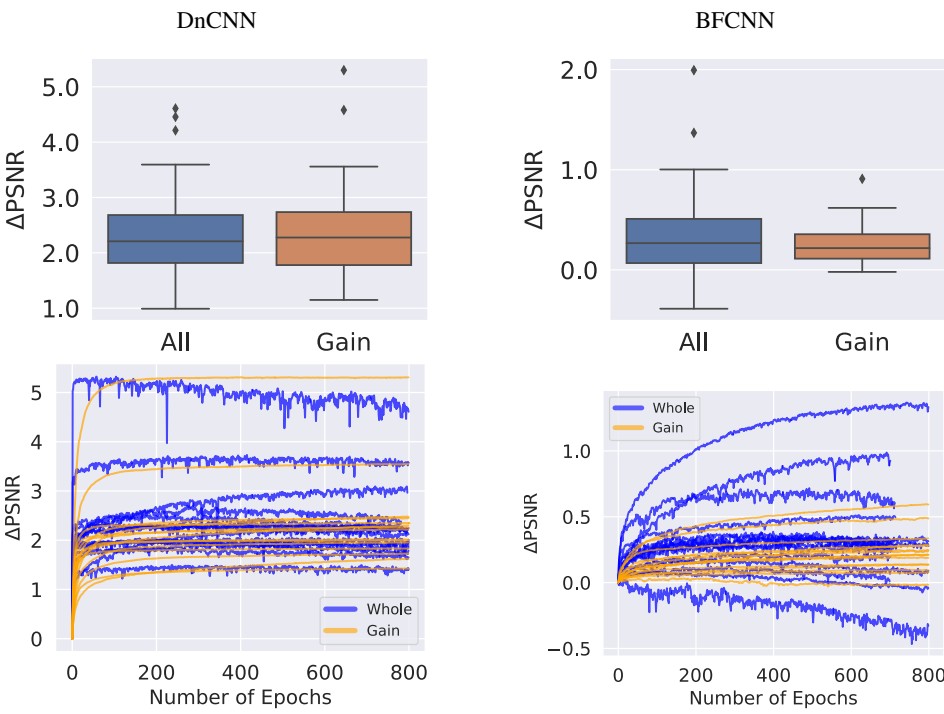

Figure 11: **Out-of-distribution noise and signal**. We compare training all parameters of the network (blue), and only the gain parameters (orange) during the adaptation process. The CNN is pre-trained on generic natural images corrupted with Gaussian noise of standard deviation $\sigma \in [0, 55]$. We apply GainTuning to adapt it to images of urban scenes (high self-similarity, hence different signal characteristics from natural images) corrupted with $\sigma = 70$ (which is outside the training range of noise). All architectures are trained using the SURE cost function.

## F.3 Out-of-distribution image

**Different Architectures**. We summarize the results using DnCNN in Table 4 in the main paper. Figures 10 show that the UNet and BFCNN architectures are also able to generalize to test data with different characteristics from the training data when adapted using GainTuning .

**Different Loss Functions**. We provide the results of evaluating the DnCNN architecture with different cost functions in Table 7.

**Comparison to baselines**. Results of comparison to LIDIA [59], a specialized architecture to perform adaptation, and Self2Self [46] a method trained exclusively on the test image is summarized in Table 6. While GainTuning outperforms LIDIA, it does not match the performance of Self2Self (see Section 7 for a discussion on this).

## F.4 Out-of-distribution noise and image

We evaluated the ability of GainTuning to adapt to test images which have different characteristics from those in the training set, and are additionally corrupted with a noise distribution that is different from the noise in the training set. Figure 11 shows that GainTuning is successful in this setting. The CNN was pre-trained on natural images corrupted with Gaussian white noise of standard deviation $\sigma \in [0, 55]$. We used GainTuning to adapt this CNN to a test set of images taken in urban setting (see Section B for a discussion on how it is different from natural images), corrupted with Gaussian noise of standard deviation $\sigma = 70$ (which is outside the training range of $[0, 55]$).

## F.5 Application to Electron Microscopy

**Comparison to pre-trained CNN**. As discussed in Section 5.4, a CNN [29] pre-trained on the simulated data fails to reconstruct the pattern of atoms faithfully. We show an additional example (Figure 15) to support this. GainTuning applied to the pre-trained CNN using the blind-spot loss correctly recovers this pattern (green box in Figure 15 (d), (e)) reconstructing the small oxygen atoms in the $CeO_2$ support. GainTuning with noise

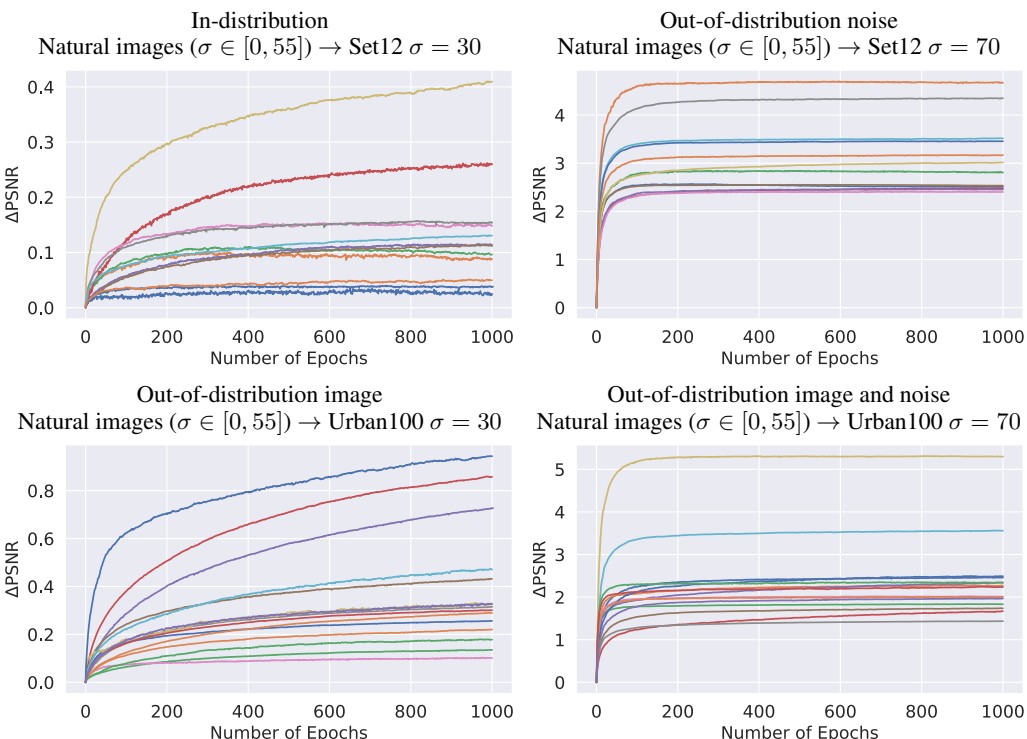

Figure 12: **GainTuning does not require early stopping**. We plot the improvement in performance achieved by GainTuning with the number of iterations. Each iteration step is a pass through 10000 random $50 \times 50$ patch extracted from the image. The performance achieved by optimizing only the gain parameters remains constant or monotonically increases with iteration, while training all parameters often overfits (see Figure 10)

resampling failed to reproduce the support pattern, probably because it is absent from the initial denoised estimate (see Figure 16).

**Comparison to baselines.** Since no ground-truth images are available for this dataset (see Section 5.4), we average 40 different acquisitions of the same underlying image to obtain an estimated reference for visual reference. We also compare GainTuning to state-of-the-art dataset based unsupervised methods, which are trained on these 40 images.

- **Blind-spot net** [29] is a CNN which is constrained to predict the intensity of a pixel as a function of the noisy pixels in its neighbourhood, without using the pixel itself. This method is competitive with the current supervised state-of-the-art CNN on photographic images. However, when applied to this dataset it produces denoised images with visible artefacts (see Figure 16). Ref. [52] shows that this may be because of the limited amount of data (40 noisy images): They trained a blind-spot net on simulated training sets of different sizes, observing that the performance on held-out data is indeed poor when the training set is small, but improves to the level of supervised approaches for large training sets.

- **Unsupervised Deep Video Denoising (UDVD)** [52] is an unsupervised method for denoising video data based on the blind-spot approach. It estimates a denoised frame using 5 consecutive noisy frames around it. Our real data consists of 40 frames acquired sequentially. UDVD produces better results than blind-spot net, but still contains visible artefacts, including missing atoms (see Figure 16). Note that UDVD uses 5 noisy images as input, and thus has more context to perform denoising than the other methods (including GainTuning ).

- **Blind-spot net with early stopping**. In Ref. [52] it is shown that early stopping based on noisy held-out data can boost the performance of blind-spot nets. Here we used 35 images for training the blind-spot net and the remaining 5 images as a held-out validation set. We chose the model parameters that minimized the mean squared error between the noisy validation images and the corresponding denoised estimates. The results (shown in Figure 16) are significantly better than those of the standard

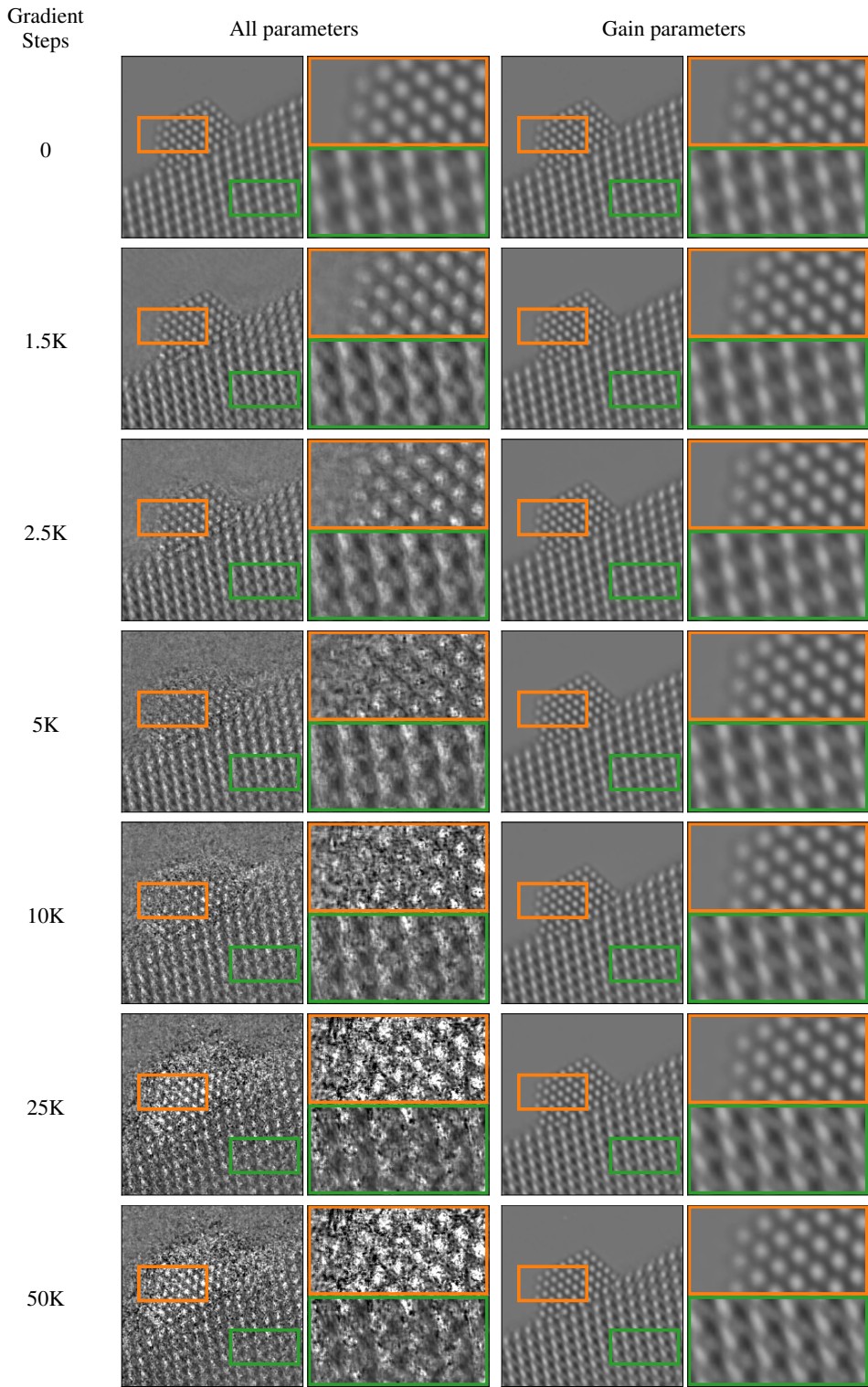

Figure 13: **GainTuning prevents overfitting in TEM data**. We compare training all parameters and only the gain parameters while adapting a CNN pre-trained on simulated TEM data to real TEM data. Training all parameters clearly overfits to the noisy image. Each gradient step is updated over two random patches of size $400 \times 400$.

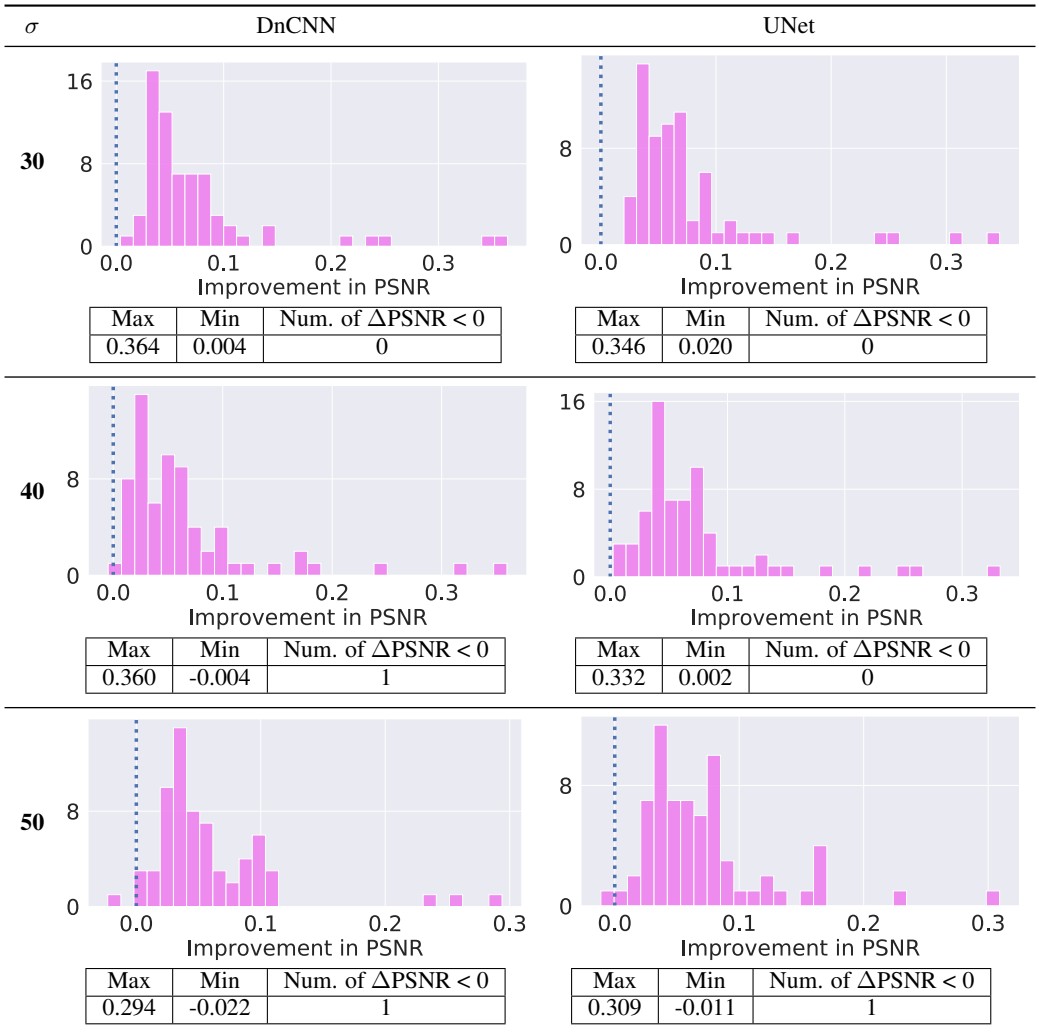

Figure 14: **Distribution of PSNR improvement on in-distribution test set**. Distribution of improvement on BSD68 dataset at noise levels $\sigma = \{30, 40, 50\}$ (in-distribution). When the network is well optimized, and the test image is in-distribution, GainTuning can sometimes degrade the performance of the network. This degradation is atypical (in this figure, there are only 3 occurrences of degradation out of 408 experiments), and very small (in this figure, the maximum degradation is 0.022)

| (a) $\zeta$ | CNN trained on Gauss. $\sigma \in [0, 55]$ | | (d) Bias-free CNN trained on Gauss. $\sigma \in [0, 55]$ | Improvement after GainTuning | |
|---|---|---|---|---|---|
| | (b) Pre-trained | (c) GainTuning | | (e) Maximum | (f) Minimum |
| 1 | 17.58 | 21.07 | 17.91 | 4.79 | 2.25 |
| 0.5 | 20.19 | 22.50 | 20.12 | 3.43 | 1.11 |
| 0.1 | 25.28 | 25.99 | 24.88 | 1.16 | 0.34 |

Table 5: **CNN trained on Gaussian noise generalizes to Poisson noise**. Results on applying GainTuning to a CNN pre-trained on additive Gaussian noise (which has spatially uniform variance) to test data corrupted by Poisson noise (where the variance depends on the underlying pixel values and is hence spatially variant). We evaluate on Poisson noise with three different scaling $\zeta$ values (a), where a larger value of $\zeta$ implies that the image is more noisy (if $x$ is a clean image, the noisy image $y$ is sampled from $\zeta\text{Pois}(x/\zeta)$ where $\text{Pois}(\lambda)$ is the PMF of Poisson distribution with parameter $\lambda$). Applying GainTuning on the CNN improves its performance (b) by a significant margin (c). GainTuning on the pre-trained CNN also outpeforms its bias-free counterpart (d), which is designed to generalize well to Gaussian noise outside the training range. The maximum improvement in PSNR (e) obtained by applying GainTuning to the pre-trained CNN (b) is substantial, and the minimum improvement in PSNR (f) is non-trivial. The CNN used here is [47] and was pre-trained on BSD400 dataset. GainTuning was performed to adapt to Set12 with Poisson noise.

| | Training Data | Test Data | DnCNN [66] | | Baselines | | |
|---|---|---|---|---|---|---|---|
| | | | | | LIDIA [59] | | S2S [46] |
| | | | Pre-trained | GainTuning | Pre-trained | Adapted | |
| (a) | Piecewise constant | Natural images | 27.31 | 28.60 | - | - | 29.21 |
| (b) | Natural images | Urban images | 28.35 | 28.79 | 28.54 | 28.71 | 29.08 |
| (c) | Natural images | Scanned documents | 30.02 | 30.73 | 30.05 | 30.23 | 30.86 |

Table 6: **GainTuning for out-of-distribution images**. GainTuning generalizes robustly when the test image has different characteristics than the training data. We demonstrate this through three different experiments. (a) GainTuning provides an average of 1.3 dB in performance while adapting a CNN trained on simulated piecewise constant dataset to natural images. This controlled setting demonstrates the capability of GainTuning to adapt from a simple simulated training set to a significantly more complex real dataset. (b) GainTuning provides an average of 0.45 dB improvement in performance when a CNN trained on natural images is adapted to a dataset of images taken in urban settings. These images display a lot of repeating structure (see Section B) and hence has different characters than generic natural images. Similarly, (c) GainTuning provides an average of 0.70 dB improvement in performance when a CNN pre-trained on natural images is adapted to images of scanned documents. While GainTuning outperforms LIDIA [59], a specialized architecture designed for adapting, it does not match the performance of Self2Self (see Section 7 for a discussion on this). As noted in Section 5.3, we did not train LIDIA for (a).

blind-spot network. However, there are still noticeable artefacts, which include missing atoms. This method is similar in spirit to GainTuning - but uses a different strategy to prevent overfitting.

- **Unsupervised Deep Video Denoising (UDVD) with early stopping**. Similar to blind-spot net, performing early stopping on UDVD using 5 held-out frames greatly improves its performance [52] (Figure 16). However, there are still noticeable artefacts in the denoised output.

## F.6 Different loss functions

GainTuning can be used in conjunction with any unsupervised denoising cost function. We explore three different choices - SURE, noise resampling, and blind-spot cost functions (see Section 4), and summarize our finding in Table 7.

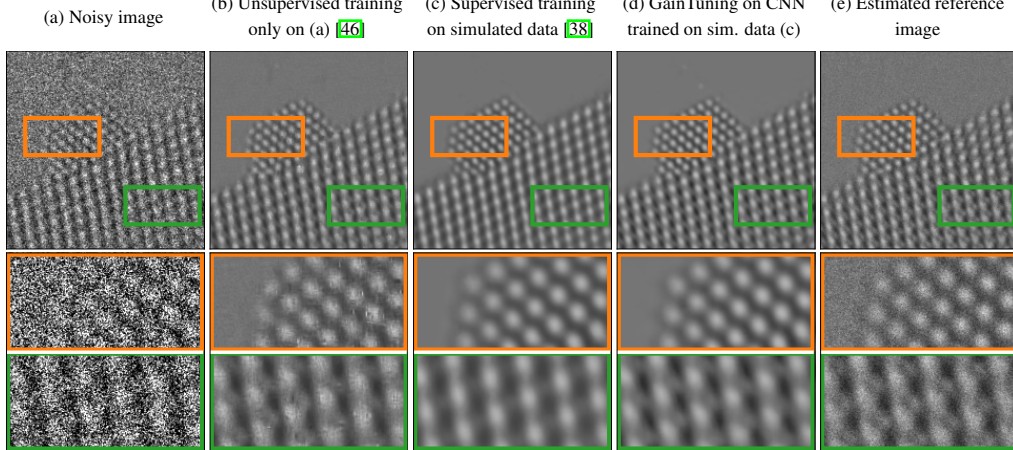

| (a) Noisy image | (b) Unsupervised training only on (a) [46] | (c) Supervised training on simulated data [38] | (d) GainTuning on CNN trained on sim. data (c) | (e) Estimated reference image |

Figure 15: **Denoising results for real-world data.** (a) An experimentally-acquired atomic-resolution transmission electron microscope image of a CeO2-supported Pt nanoparticle. The image has a very low signal to noise ratio (PSNR of $\approx 3dB$). (b) Denoised image obtained using Self2Self [46], which contains significant artefacts. (c) Denoised image obtained via a CNN trained on a simulated dataset, where the pattern of the supporting atoms is not recovered faithfully (third row). (d) Denoised image obtained by adapting the CNN in (c) to the noisy test image in (a) using GainTuning. Both the nanoparticle and the support are recovered without artefacts. (e) Reference image, estimated by averaging 40 different noisy images of the same nanoparticle. See Figure 2 for an additional example.

SURE loss outperforms other choices in most experiments. Noise resampling has comparable performance to SURE when the test data is in-distribution, or when it is corrupted with out-of-distribution noise. However, noise resampling generally under-performs SURE when the test images have different features from the training images. A possible explanation for this is that noise resampling relies on the initial denoised image to fine-tune and, therefore, it may not be able to exploit features which are not present in the initial estimate. In contrast, the SURE cost function is computed on the noisy test image itself, thereby enabling it to adapt to features that the pre-trained network may be agnostic to.

Finally, adapting using blind-spot cost function often under-performs both SURE and noise resampling. The difference in performance is reduced at higher noise levels (see also Section 5.4 where we use blind-spot cost function for experiments with real TEM data with very high noise). The reason for this could be that at higher noise levels, the information contained in a single pixel becomes less relevant for computing the corresponding denoised estimate (in fact, the regularization penalty on "self pixel' for SURE cost function (Section 4) increases as the noise level increases). Therefore, the loss of performance incurred by the blind-spot cost function is diminished. At lower noise levels (particularly when the images are in-distribution), adapting using blind-spot cost function will force the pre-trained network to give up using the "self pixel", which results in a degraded performance. An alternative to adapting a generic pre-trained network using blind-spot architecture is to use a CNN that is architecturally constrained to include a blind-spot. In Table 8, we show that adapting such a CNN using blind-spot loss improves the performance its performance. However, the overall performance of this architecture is in general lower than the networks which also use the "self pixel". We refer interested readers to Ref. [29, 28, 62] for approaches to incorporate the noisy pixel into the denoised estimate.

# G    Analysis

## G.1    What kind of images benefit the most from adaptive denoising?

We sort images by the improvement in performance (PSNR) achieved with GainTuning. We observe that the ordering of images is similar for different models and cost functions (See Figure 17), implying that the improvement in performance is mostly dependent on the image content. The images with largest improvement typically contain repeated patterns and are more structured. Repetition of patterns effectively provides multiple samples from which the unsupervised refinement can benefit.

| | | Pre-training | SURE | Noise resampling | Blind-spot (Noise2Self [3]) |
|---|---|---|---|---|---|
| | | | | GainTuning with | |
| in distribution | Set12 | 29.52 | 29.62 | **29.63** | 29.50 |
| | BSD68 | 28.39 | **28.46** | 28.40 | 28.36 |
| out-of-distribution noise | Set12 | 18.48 | **24.57** | 24.11 | 22.93 |
| | BSD68 | 18.72 | **24.14** | 23.65 | 22.50 |
| out-of-distribution image | Piecewise constant → Natural images | 27.31 | **28.60** | 28.29 | 27.39 |
| | Natural images → Urban100 | 28.35 | **28.79** | **28.79** | 28.29 |
| | Natural images → Scanned documents | 30.02 | **30.73** | 30.57 | 29.23 |

Table 7: **Different loss functions for GainTuning**. Comparison of the performance of GainTuning when used in conjunction with three different loss functions. SURE loss outperforms other choices in most experiments. Noise resampling has comparable performance to SURE when the test data is in-distribution, or when it is corrupted with out-of-distribution noise. However, noise resampling generally under-performs SURE when the test images have different features from the training images. This maybe because such features are absent from the initial denoised estimate (see Section 4 for a description of the different loss functions). Finally, optimizing using blind-spot cost functions often under-performs both SURE and noise resampling, but the difference in performance is reduced as the test noise increases (see also Section 5.4 where we use blind-spot cost function for experiments with real TEM data with very high noise). This may be because, at lower noise levels, the information contained in a pixel is often crucially important to compute its denoised estimate, and blind-spot cost function ignores this information (see Section 4). Here, we implemented blind-spot cost function through masking [3], see Table 8 for results where the implemented blind-spot cost function as an architectural constraint [29].

| | in-distribution | | out-of-distribution image | |
|---|---|---|---|---|
| | Set12 | BSD68 | Urban100 (urban scenes) | IUPR (scanned documents) |
| Pre-trained | 27.92 | 26.47 | 26.59 | 28.25 |
| GainTuning | 27.92 | 26.61 | 26.85 | 28.40 |

Table 8: **GainTuning using architecturally constrained blind-spot cost function**. We perform GainTuning using blindspot network [29] which is architecturally constrained to estimate a denoised pixel exclusively from its neighbouring pixels (excluding the pixel itself). The network was pre-trained on generic natural images corrupted with Gaussian noise of standard deviation $\sigma \in [0, 55]$. Performing GainTuning on this always increases its performance, unlike GainTuning on a generic architecture trained with supervision and adapted using blind-spot loss implemented via masking. However, note the overall performance of this architecture is in general lower than the networks which also use the "self pixel". We refer interested readers to Ref. [29, 28, 62] for approaches to incorporate the information in noisy pixel back into the denoised output, thus potentially improving the performance. Our blind-spot architecture generalizes robustly to out-of-distribution noise (since it is bias-free [37]), and therefore we do not include an out-of-distribution noise comparison in this table.

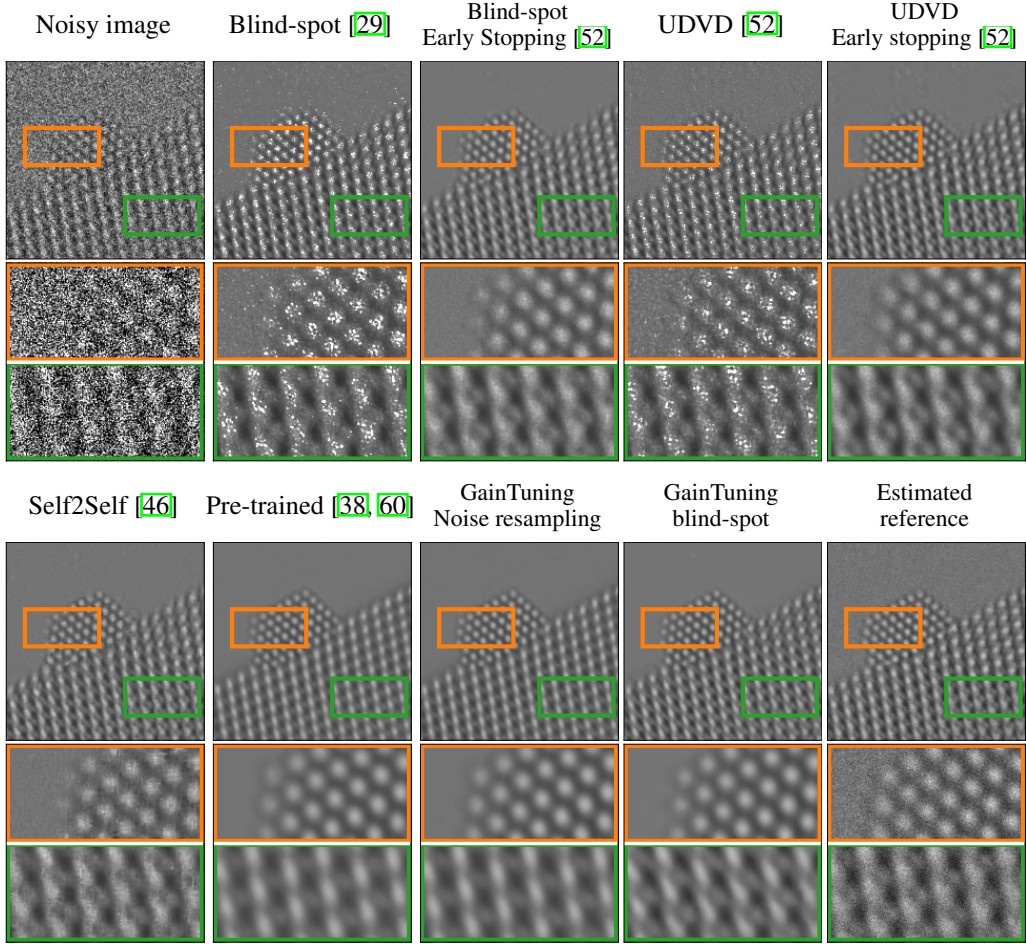

Figure 16: **Comparison with baselines for electron microscopy**. GainTuning clearly outperforms Self2Self, which is trained exclusively on the real data. The denoised image from Self2Self shows missing atoms and substantial artefacts (see Figure 15 for another example). We also compare GainTuning dataset to blind-spot methods using the 40 test frames [29, 52]. GainTuning clearly outperforms these methods.

## G.2 Generalization via GainTuning

We investigate the generalization capability of GainTuning. We observe that a CNN adapted to a particular image via GainTuning generalizes effectively to other similar images. Figure 18 shows that GainTuning can achieve generalization to images that are similar to the test image used for adaptation on two examples: (1) adapting a network to an image of a scanned document generalizes to other scanned documents, and (2) adapting a a network to an image with out-of-distribution noise generalizes to other images with similar noise statistics.

## G.3 How does GainTuning adapt to out-of-distribution noise?

Let $y \in \mathbb{R}^N$ be a noisy image processed by a CNN. Using the first-order Taylor approximation, the function $f : \mathbb{R}^N \to \mathbb{R}^N$ computed by a denoising CNN may be expressed as an affine function

$$f(z) = f(y) + A_y(z - y) = A_y z + b_y, \tag{10}$$

where $A_y \in \mathbb{R}^{N \times N}$ is the Jacobian of $f(\cdot)$ evaluated at input $y$, and $b_y \in \mathbb{R}^N$ represents the *net bias*. In [37], it was shown that the bias tends to be small for CNNs trained to denoise natural images corrupted by additive Gaussian noise, but is a primary cause of failures to generalize to noise levels not encountered during training. Figure 18 shows that GainTuning reduces the net bias of CNN, facilitating the generalization to new noise levels.

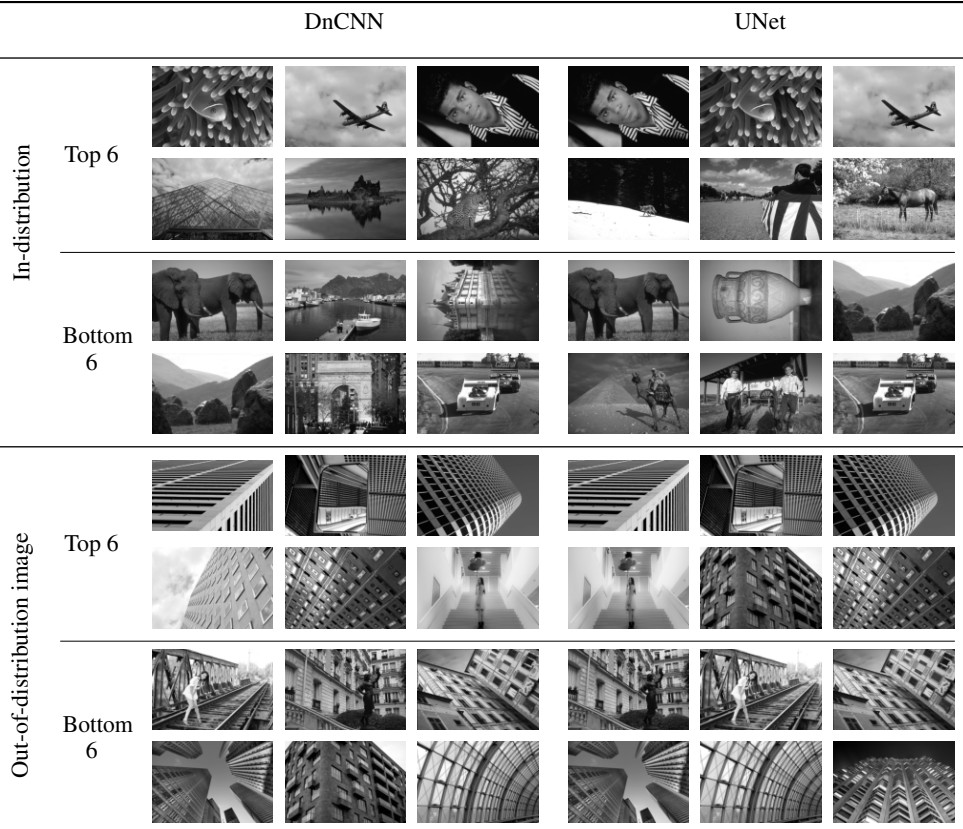

Figure 17: **What kind of images benefit the most from adaptive denoising?** We visualize the images which achieve the top 6 and bottom 6 (left top to the right bottom of each grid) improvement in performance (in PSNR) after performing GainTuninġImages with the largest improvement in performance often have highly repetitive patterns or large regions with constant intensity. Images with least improvement in performance tend to have more heterogeneous structure. Note that, in general, the distribution of improvements in performance is often skewed towards the images with minimal improvement in performance (See Figures 3, 4, and 14).

## G.4   How does GainTuning adapt to out-of-distribution images?

In order to understand how GainTuning adapt to out-of-distribution images, we examine the adaptation of a CNN pre-trained on piecewise constant to natural images. Piecewise constant images have large areas with constant intensities, therefore, CNNs trained on these images tends to average over large areas. This is true even when the test image contains detailed structures. We verify this by forming the affine approximation of the network (eq. 10) and visualizing the equivalent linear filter [37], as explained below:

Let $y \in \mathbb{R}^N$ be a noisy image processed by a CNN. We process the test image using a Bias-Free CNN [37] so that the net bias $b_y$ is zero in its first-order Taylor decomposition (10). When $b_y = 0$, (10) implies that the $i$th pixel of the output image is computed as an inner product between the $i$th row of $A_y$, denoted $a_y(i)$, and the input image:

$$f(y)(i) = \sum_{j=1}^{N} A_y(i,j)y(j) = a_y(i)^T y. \tag{11}$$

The vectors $a_y(i)$ can be interpreted as *adaptive filters* that produce an estimate of the denoised pixel via a weighted average of noisy pixels. As shown in Figure 5 the denoised output of CNN pre-trained on piece wise constant images is over-smoothed and the filters average over larger areas. After GainTuning the model learns to preserve the fine features much better, which is reflected in the equivalent filters.

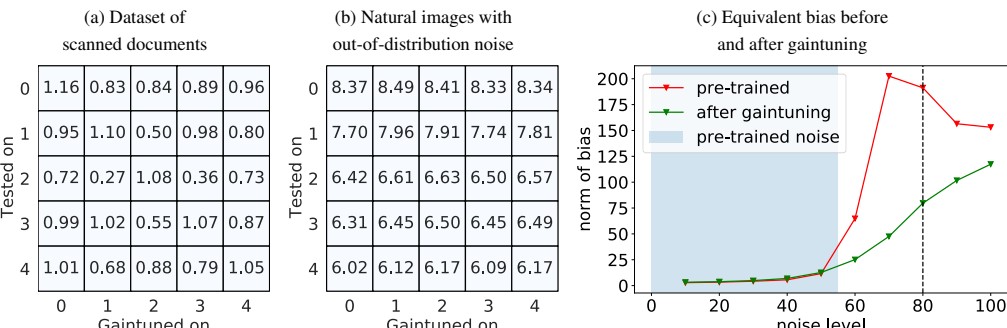

Figure 18: **Analysis of GainTuning**. GainTuning can achieve generalization to images that are similar to the test image used for adaptation. We show this through two examples: (a) adapting a network to an image of a scanned document generalizes to other scanned documents, and (b) adapting a a network to an image with out-of-distribution noise generalizes to other images with similar noise statistics. The $(i, j)^{\text{th}}$ entry of the matrix in (a) and (b) represents the improvement in performance (measured in PNSR) when a CNN GainTuned on image $j$ is used to denoise image $i$. We use 5 images with the largest improvement in performance across the dataset for (a) and (b). Finally, (c) shows that generalization to noise levels outside the training range is enabled by reducing the *equivalent bias* of the pre-trained CNN (see equation (10)).