# OpenReview forum: "Adaptive Denoising via GainTuning"
_NeurIPS.cc/2021/Conference — NeurIPS 2021 Poster_

### Official Review · Reviewer_sYYW · 2021-07-13

**Rating:** 6
**Confidence:** 4

**Summary:**

In this work the authors propose a new approach to adapt pre-trained denoising model to new test scenarios not seen at train time both in terms of noise distribution and image content. The proposed solution is fairly simple and straightforward: a gain parameter is estimated per channel per layer for the denoising neural network model at test time for each image. These gain parameters are estimated using the following unsupervised cost functions on the test image: Blind-spot loss, SURE loss, Noise Resampling [56].

**Limitations And Societal Impact:**

The authors discuss the limitations of the model and point to the restricted optimization space which can limit the possible improvements. However I think the experiments did not investigate enough more complex noise models and it would be important to address such issues.

**Main Review:**

**Originality**

The proposed idea is interesting and a relatively simple way to manipulate the output from a denoising model. There is a relationship with feature modulation techniques such as AdaFM and “Cfsnet:  Toward  a  controllable  featurespace for image restoration” that is not sufficiently discussed.
In addition to this “Interactive Multi-Dimension Modulation with Dynamic Controllable Residual Learning for Image Restoration” (ECCV 2020) should be discussed. It adopts similar feature modulation to target multiple degradation.

The optimization process is also related to some blind super resolution methods such as:

-Blind Super-Resolution With Iterative Kernel Correction

-Blind image super-resolution with spatially variant degradations,

In the case of denoising unsupervised cost functions could be directly used and a separate network is not needed, which allows to potentially better target “out of distribution” noise levels.


**Quality**

The method achieves good results and the authors demonstrate a wide variety of scenarios where the proposed model performs better than pre-trained models and blind denoising methods overfitted to the test data. There are however a few elements that needs explanations and more details:

-The authors only demonstrate results on gray scale images. Is there a reason to not investigate denoising for color images?

-The proposed noise distribution is relatively simple (Gaussian). More complex noise distribution should be investigated in particular signal dependent models? It is not clear if the model would be adapted to those situations.

-Still related to the noise adaptation, it is not clear if it would be possible to have locally different denoising gains?

-Authors should discuss the link with some existing denoising method that takes into account the noise level as conditioning, such as “FastDVDnet: Towards Real-Time Deep Video Denoising Without FlowEstimation”. In addition, in this case a similar optimization strategy could be used to estimate noise levels; This seems like a reasonable baseline to demonstrate the interest of the gain.

**Clarity**

The paper is clear and easy to follow. All the experiments are sufficiently well described and the supplemental material provides even more details.


**Significance**

The paper presents an interesting idea and denoising is one of the most common problems, however a few questions need to be answered to appropriately judge the significance.

**After rebuttal**

The authors answered most of my questions and concerns in their rebuttal. I would insist they include discussion with respect to the different papers that were missing in the first version. If possible, I would also advise including experimental comparisons against methods like FastDVDnet.

**Time Spent Reviewing:**

4

---

> ### Author Response · Authors · 2021-08-10
> **Response to ReviewersYYW.**
>
> Thank you for your thoughtful and detailed review. Specific responses are below:
>
> >  There is a relationship with feature modulation techniques such as AdaFM and … The optimization process is also related to some blind super resolution methods such …
>
> Thank you for the references. We will expand our related work section to include these papers as well. AdaFM ([20] in our references) learns modulation parameters for intermediate layers at different degradation levels (using supervision), and interpolates between these values to apply the network at a new degradation level. In contrast, GainTuning optimizes the gain parameters using only the test image to adapt to new degradation levels (or noise types) and new signal types. CFSNet and He. et. al, ECCV 2020 also modulates the input channels, but the modulates are incorporated into the network architecture and learned during training stage, unlike in GainTuning where any pre-trained denoiser can be adapted to a given test image dynamically at test time. We will also include the references on super-resolution and comment on possible implications of GainTuning in this setting.
>
> > The authors only demonstrate results on gray scale images. Is there a reason to not investigate denoising for color images?
>
> We did this experiment, and find that GainTuning is effective on color images. We’ve tested DnCNN trained on color images with noise in the range $\sigma \in [0, 55]$ on (1) in-distribution data: natural images with with Gaussian noise of $\sigma=30$, (2) out-of-distribution noise images: natural images with Gaussian noise ($\sigma=100$), and (3) out-of-distribution signal: images of urban scenes corrupted with Gaussian noise ($\sigma = 30$).  The results are summarized below:
>
> |                                | Pre-trained | GainTuning | Improvement in PSNR |
> |--------------------------------|-------------|------------|---------------------|
> | (1) in-distribution            | 29.85       | 30.07      | 0.22                |
> | (2) out-of-distribution noise  | 14.80       | 23.66      | 8.86                |
> | (3) out-of-distribution signal | 29.52       | 29.85      | 0.33                |
>
>
> We provide visual examples of out-of-distribution noise generalization with GainTuning [here](https://drive.google.com/file/d/18gCEaYedFVCrN9orVnGEe2cHotrjo2nb/view?usp=sharing). We will add these results and the example to the revised version of the manuscript.
>
> > The proposed noise distribution is relatively simple (Gaussian). More complex noise distribution should be investigated in particular signal dependent models? It is not clear if the model would be adapted to those situations.
>
> GainTuning is a general framework which can handle data shifts including out-of-distribution noise, signal or both noise and signal, but we agree that adding more complex noise distributions is of interest. To this end, we have tested GainTuning a CNN pre-trained on additive Gaussian noise (which has fixed constant variance) to test data corrupted by Poisson noise (where the variance depends on the underlying pixel values and is hence spatially variant). GainTuning produces a substantial improvement in the denoising performance of the CNN (around 3.5 dB on Set12. See [table here](https://drive.google.com/file/d/1sKwPh6_rzpU2fMMjlhGVenpSL5mGboXe/view?usp=sharing) ).  These results are obtained using GainTuning based on the blind-spot loss on a UNet architecture [47] pre-trained on natural images with Gaussian noise of standard deviation between 0 and 55. We will include a more comprehensive version of this experiment in the revised version of the paper.
>
> > Still related to the noise adaptation, it is not clear if it would be possible to have locally different denoising gains?
>
> This is a very interesting topic we would like to explore. Spatially local gain control has been used in some applications including computational neuroscience [5], density estimation [1] and compression [2]. Incorporating a locally adaptive mechanism on the gains may be effective in certain situations. We will mention this as an interesting research direction in the discussion.
>
> > Authors should discuss the link with some existing denoising method that takes into account the noise level as conditioning, such as “FastDVDnet: Towards Real-Time Deep Video Denoising Without FlowEstimation”. In addition, in this case a similar optimization strategy could be used to estimate noise levels; This seems like a reasonable baseline to demonstrate the interest of the gain.
>
> This is another interesting suggestion. We will discuss the strategy in FastDVDnet of conditioning on the noise level. In terms of baseline, for the specific case of additive Gaussian noise we chose the baselines that are reported to perform best in the literature (training on all noise levels beforehand and removing additive biases from the architecture). GainTuning is able to approximate the performance of these specialized baselines just from the test image, without retraining on the training set or architectural modifications. Moreover, as demonstrated by our experiments on electron microscopy and by the additional experiment on adaptation to Poisson noise, GainTuning is also able to generalize to different noise distributions
>
>
> > I think the experiments did not investigate enough more complex noise models and it would be important to address such issues.
>
> We completely agree that it is important to consider complex noise models.  We would like to highlight that we have included results on a real-world application to transmission electron microscope (Section 5.4). These data are corrupted by spatially-variant shot noise, as well as other noise processes such as readout noise and dark current. Our results (see Figures 2, SM9, SM10) show that GainTuning is very effective in denoising these real-world data with complex noise, clearly outperforming alternative approaches. Further, we have also added a controlled experiment where the noise behavior is qualitatively different to the noise model used for pretraining: we have applied GainTuning to a CNN pre-trained on additive Gaussian noise (which has fixed constant variance) to test data corrupted by Poisson noise (where the variance depends on the underlying pixel values and is hence spatially variant and more complex).

---

> ### Author Response · Authors · 2021-09-14
> **Thank you for responding to our rebuttal.**
>
> Thank you for responding to our rebuttal.
> + We will discuss connections with additional references in our final version - we have included a summary of this discussion in the rebuttal.
> + FastDVDnet is designed for video denoising (unlike our paper which concentrates on image denoising), but thank you for this suggestion. We think employing GainTuning for video denoising will be an interesting research direction, and will include some preliminary experiments in the final version.

---

### Official Review · Reviewer_YFNi · 2021-07-14

**Rating:** 6
**Confidence:** 4

**Summary:**

This paper proposes a new approach for image denoising, which combines pretraining a neural network on a dataset, and then fine tuning it on the noisy test image to be denoised. The key new feature is to tune only a small set of parameters, called gains, one for each channel within each layer in the network. These gains multiply the corresponding channels.
The benefit of the approach is illustrated in the task of removing white Gaussian noise. The method is shown to improve performance both in the standard denoising setting, and in cases where the noise distribution or the image distribution (or both) in the test set differ from the train set.

**Limitations And Societal Impact:**

The authors have addressed limitations and societal impact.

**Main Review:**

The idea of combining external supervision with internal learning, is not new, and has been already explored in several contexts, including super-resolution and denoising. However, the particular way proposed in this paper seems to be new, and is quite interesting. It makes a lot of sense to restrict the degrees of freedom in the fine-tuning stage, and doing so by controlling channel gains is shown to lead to good results. However, I do have several concerns.

*Generalization to out-of-distribution noise*

A major claim in the paper is that the proposed method generalizes well to out-of-distribution noise. But the method is still inferior to [35] in this task (Fig. 4). The authors explain this by referring to [35] as "specifically designed to generalize to out-of-distribution noise". But all that [35] really do is remove the additive biases within all the layers of the network. They showed that this turns any denoising network into one that generalizes well to different noise-levels. What's bothering here is that the method in the current manuscript also uses no biases, as can be learned from the supplementary (SM 1.1). So the authors can't blame the poorer performance w.r.t. [35], on any specialized architecture. It seems to me that the authors' experiments prove the exact opposite of what they claim. Namely, GainTuning rather *impairs* the ability of a bias-free network to generalize well to out-of-distribution noise. Indeed, [35], which uses a bias free network but without GainTuning, performs better than GainTuning.

*Experiments*

- It's not clear whether the UNet architecture also uses no biases. This should be stated and emphasized in the paper.

- It's also not clear which architecture was used in which experiment (DnCNN or UNet or something else). This is very important for assessing the comparisons.

-The three fine-tuning losses: A whole section (Sec. 4) is devoted to explaining about three possible losses for fine-tuning, but no comparison between them is provided in the main text (only in the supp.).  Line 189 refers to Table 2 for such a comparison, but no such table exists. Also, it's not clear which loss was eventually used in each experiment (for example in Figs. 3-5).

- Visualization in Fig. 5: The visualization of the filter coeff. in Fig. 5 should be explained. I understand this is taken from [35], which only makes sense for bias-free networks, but it should be explained in the current paper for completeness.


*Exposition*

-SURE: The terminology used is a bit inconsistent with the original concept. SURE refers to what's inside the second expectation in (5), namely it's without the expectation. And this is also the loss that's being optimized in practice in this paper (without the expectation), because there is only a single y. Stein's theorem shows that the expectation over y of SURE is the expectation over y of $||x-f(y)||^2$, for any given $x$.

-Eq. (7): Extra left parentheses.

-Line 201: Missing ref. in "(Figure 4 and)"


**---------------- Post Rebuttal ----------------**
My main concern had to do with the biases in the networks. This has been clarified in the rebuttal. I therefore recommend acceptance.


**Time Spent Reviewing:**

5

---

> ### Author Response · Authors · 2021-08-10
> **Response to Reviewer YFNi.**
>
> Thank you for the thoughtful and detailed comments. We are glad that you found our method novel and interesting. Specific responses are below:
>
> >   the method is still inferior to [35] in this task.  But all that [35] really do is remove the additive biases within all the layers of the network.
>
> This was not explained clearly enough in the paper. Ref [35] shows that **removing all additive terms** in a denoising CNN yields a network that generalizes much better - that is, when it is **trained from scratch** on a narrow range of noise levels, it generalizes to other noise levels at test time.  In our experiments, we apply GainTuning to a CNN with additive (bias) terms, which would ordinarily *not* generalize well  to new noise levels. GainTuning enables generalization, producing a substantial performance gain of 6dB as reported in Fig 3. This is a controlled setting, which demonstrates that GainTuning is able to enhance a generic CNN to approximately match the state-of-the-art.  Although the performance of GainTuning is slightly lower than the bias-free CNN, it is achieved based only on fine-tuning on the test image, without requiring modifications in architecture or re-training on a large training set. This is important for more realistic settings, where specialized architectures may not be available (and the noise properties may not even be known beforehand). Our results on electron microscopy illustrate the potential of GainTuning in such situations (we have also added an experiment on generalization from Gaussian to Poisson noise, see [table here](https://drive.google.com/file/d/1sKwPh6_rzpU2fMMjlhGVenpSL5mGboXe/view?usp=sharing)).
>
> > What's bothering here is that the method in the current manuscript also uses no biases, as can be learned from the supplementary (SM 1.1).
>
> All networks used for the out-of-distribution noise experiments have biases and do not generalize well, as shown in the “pretrained” column of Figure 3. SM 1.1 describes DnCNN and its bias-free version. We apologize for the confusion, and will edit the text to make this clearer.
>
>
> > GainTuning rather impairs the ability of a bias-free network to generalize well to out-of-distribution noise.
>
> This is a misunderstanding (due to ambiguities in our text).  GainTuning substantially improves the performance of CNNs with biases on out-of-distribution noise. GainTuning can also be applied to bias-free CNNs (which already generalize to noise levels outside their training range) to slightly improve their performance. For example, BFCNN (bias-free version of DnCNN) pre-trained for Gaussian noise removal on $\sigma \in [0, 55]$ achieves a performance of 25.59 dB on Set12 at $\sigma=70$. After applying GainTuning, the performance increases to 25.64 dB (this is similar to the performance of GainTuning on BFCNN with in-distribution test data, see Table SM 2). We will include these results in the paper.
>
> > It's not clear whether the UNet architecture also uses no biases. It's also not clear which architecture was used in which experiment (DnCNN or UNet or something else). Also, it's not clear which loss was eventually used in each experiment (for example in Figs. 3-5).
>
> We apologize for the confusion, and will update all the figure captions to include the details of the architecture and loss function. The UNet architecture uses bias (except BFCNN, all architectures have bias). For all controlled experiments in the main paper, we used DnCNN and SURE loss (in Fig 3 we used both DnCNN and UNet, see lines 187-188, 232-233), but we report results with other architectures and loss functions in the supplementary material (Section SM 6).
>
> >  A whole section (Sec. 4) is devoted to explaining about three possible losses for fine-tuning, but no comparison between them is provided in the main text (only in the supp.). Line 189 refers to Table 2 for such a comparison, but no such table exists.
>
> We agree that it would be good to have this in the main paper.  We had moved it to the supplementary material due to space constraints, but will try to move it back if possible. The missing Table 2 in main text is a typo, and should refer instead to Table SM 5 in the supplementary material.
>
> > Visualization in Fig. 5 ... only makes sense for bias-free networks, …
>
> We agree (it is stated lines 311-312 in the supplementary material).  We will make this explicit in the caption and the main text.
>
> > Exposition
>
> Thank you for reading our paper carefully. We appreciate your comments, and will address all of them in the revised version.

---

### Official Review · Reviewer_uD7E · 2021-07-14

**Rating:** 6
**Confidence:** 4

**Summary:**

The authors present a novel training method for denoising that combines pre-training with unsupervised/selfsupervised finetuning on the test image.
To avoid overfitting during finetuning , the authors adjust not the weights of the network but only single scalar variables for each feature channel in each layer of the network.
They evaluate their method on different on different datasets and with different self-supervised/unsupervised loss functions and achieve remarkable results.

**Ethical Concerns:**

I do not have any ethical concerns regarding the paper.

**Limitations And Societal Impact:**

I do not see any negative societal impacts. Regarding limitations, please see Main Review.

**Main Review:**

========================
Strengths
========================

Adaptation to test data that is different from the training set is an important problem.

The results are convincing.

I like the fact that the method is viewed as a general approach, and applied with different established network architectures and loss functions.
This is in contrast to many papers that present a new training scheme coupled with a modified architecture, making it difficult to disentangle the advantages of each component.

I appreciate the extensive experiments documented in the supplementary material.

The paper is well written and easy to follow.

========================
Weaknesses
========================

There are some open questions for me regarding the assessment of the potential and limitations of the method:
In line 277 the authors write:
"GainTuning is more beneficial for images with highly repetitive patterns".
Yet, in the Limitations sections they state that for images containing repetitive structure, finetuning all parameters outperforms the method.
Can the authors please clarify this?
It makes intuitive sense that images with repeating structures are less prone to overfitting. Shouldn't Gain tuning be most beneficial for images that do not contain repeating structures?

It is important for the paper to show that overfitting is real problem.
Supplementary Figures SM2 and SM3 discuss this, but it looks like overfitting mainly seems to be a problem of SURE and occurs to a much smaller extent with Noise Resampling. Why is this the case? Why is Blind spot denoising missing from the analysis in Supplementary Figures SM2 and SM3?


========================
Minor Points
========================

Table 2 is referenced in line 189, but I cannot seem to find it in the paper.
It would actually be interesting to have these results.

I think [1] is highly related and could be discussed.
[1]: Wang, Yina, et al. "Image denoising for fluorescence microscopy by self-supervised transfer learning." bioRxiv (2021).


========================
Final Recommendation
========================

I think the paper presents a solid contribution to an important topic.
Given an adequate response and clarifications regarding the mentioned weaknesses I am happy to support it.


========================
Post Rebuttal
========================

I am happy with the answers provided by the authors and I recommend accepting the paper.
I believe my questions regarding overfitting and the performance of SURE and Noise Resampling are sufficiently answered and I very much appreciate the additional Figure, which I hope can be included in the supplementary material.


**Time Spent Reviewing:**

5

---

> ### Author Response · Authors · 2021-08-10
> **Response to Reviewer uD7E.**
>
>
> Thank you for your encouraging review and thoughtful questions. Specific responses are below.
>
> > It makes intuitive sense that images with repeating structures are less prone to overfitting. Shouldn't Gain tuning be most beneficial for images that do not contain repeating structures?
>
> Yes, your intuition is correct. This is a subtle point that was not sufficiently clear in the paper. The improvement in performance achieved by GainTuning is larger if the test image contains more repeating structure (this is what we meant by “beneficial” in line 277). At the same time, as you note, when there is repeating structure we are less prone to overfitting, so there is less advantage in using GainTuning with respect to fine-tuning all the parameters. For images with less structure, fine-tuning all parameters often degrades performance noticeably, while GainTuning always increases or maintains performance (see Section SM5). In this sense, GainTuning is relatively more beneficial for images with less structure! We will make sure that this is clear in the revised version.
>
> > overfitting mainly seems to be a problem of SURE and occurs to a much smaller extent with Noise Resampling. Why is this the case?
>
> This is a great question. SURE and the blind-spot cost function both force the network to approximate each noisy pixel in the test image, while preventing the network from using the pixel itself to prevent overfitting (see lines 145-156). When all parameters in the network are fine-tuned using just a single image, memorization can occur due to the small number of examples with respect to the parameters. For example, DnCNN has 600K parameters, and a 256x256 image has 65K pixels. In contrast, noise resampling adds synthetic noise to the initial denoised image to produce an arbitrarily large number of training examples. This prevents overfitting the noisy test image to some extent (GainTuning is still advantageous). However, it also means that the final solution cannot be too far from the initial denoised image. The original noisy image is not available to the network, which limits the ability of the model to learn structures that are not present in the initial denoised image (for example in Fig SM 10 noise resampling cannot reconstruct the bulk pattern in the electron microscopy example). We will elaborate on this in the revised version of the paper.
>
> > Why is Blind spot denoising missing from the analysis in Supplementary Figures SM2 and SM3?
>
> We agree that including blind-spot denoising would be useful there. The [figure here](https://drive.google.com/file/d/1cNOkiA1akctYMTMaufUTmndmIdV9QzW1/view?usp=sharing) shows the box plot of the improvement in performance, and a scatter plot of the performance before and after fine-tuning when updating all (blue) parameters and just the gain (orange) parameters, for in-distribution signal/noise, out-of-distribution noise and out-of-distribution signal (see the caption of the image for details about the experiment). Updating all parameters shows significant overfitting. In contrast GainTuning consistently improves the denoising performance.
>
> >  Table 2 is referenced in line 189, but I cannot seem to find it in the paper.
>
> Our apology - this was a typo, and should refer instead to Table SM5 in the supplementary material.
>
> >  I think [1] is highly related and could be discussed. [1]: Wang, Yina, et al. "Image denoising for fluorescence microscopy by self-supervised transfer learning." bioRxiv (2021).
>
> Thank you for the reference. Wang, Yina et, al, demonstrates good results in fluorescence microscopy, using a fine-tuning method that is similar in spirit but different in detail. Specifically, in contrast to our setting where only a single noisy image is available for fine-tuning, Wang, Yina et al. uses a dataset of noisy images to perform fine-tuning, potentially reducing the risk of overfitting. We will include a discussion of this work in the revised version.

---

> > ### Comment · Reviewer_uD7E · 2021-08-31
> > **I am happy with the answers provided by the authors and I recommend accepting the paper.**
> >
> > I believe my questions regarding overfitting and the performance of SURE and Noise Resampling are sufficiently answered and I very much appreciate the additional Figure, which I hope can be included in the supplementary material.

---

> > > ### Author Response · Authors · 2021-09-01
> > > **Thank you!**
> > >
> > > Thank you for the encouraging response. We will include the additional figure and experiments performed during rebuttal in our final submission.

---

### Official Review · Reviewer_aT3f · 2021-07-18

**Rating:** 5
**Confidence:** 4

**Summary:**

In this paper a new method is proposed to selectively fine-tune the existing neural network for denoising using the test image. In order to achieve this the authors propose to use additional trainable gain parameters for each channel in each layer and optimize these parameters during fine-tuning. The authors found that by doing this the network is well adapted to the target test images even if the noise level and image type are novel to the original neural network.

**Ethics Review Area:**

["I don’t know"]

**Limitations And Societal Impact:**

There should be no potential negative societal impact of their work

**Main Review:**

In this paper a new method is proposed to selectively fine-tune the existing neural network for denoising using the test image. In order to achieve this the authors propose to use additional trainable gain parameters for each channel in each layer and optimize these parameters during fine-tuning. The authors found that by doing this the network is well adapted to the target test images even if the noise level and image type are novel to the original neural network.

The idea of the paper and the application of such method in the test stage are interesting. The analysis of the proposed method in the paper is relevant. The experiments seem to supported the claimed contribution. While there're merits, I have several concerns of this paper. First of all, the authors claim that the method works when the noise level and image type are novel to the original network. However, in practice the situation in the test images could be much more complex. It is not uncommon that the test images' noise pattern is novel. It is also not uncommon that the noise exists in the images are not uniform. This is particularly common when the image is not in the raw domain (after ISP's nonlinear operators).  I wonder if the method would still work in these common and important situations. If the method has difficulty in deal with these practical situations, the value of using such method against any novel test image during test time is limited. Secondly, because there're limited number of the parameters to tune during test time, I suspect the original network needs to be big and powerful so that it may offer sufficient flexibility to the proposed approach. This may bring significant difficulty to network compression and quantization if one was to deploy such technology in real life. I believe this issue is relevant to such an application oriented paper.

**Time Spent Reviewing:**

3

---

> ### Author Response · Authors · 2021-08-10
> **Response to Reviewer aT3f.**
>
> Thank you for your valuable feedback, which will improve our paper. We are glad that you find our idea interesting, our proposed method and analysis relevant, and our claims well supported by experiments. We believe that our results on electron microscopy (as well as an additional experiment on generalization from Gaussian to Poisson noise) showcase the potential of the proposed method for practical settings with realistic noise. We respectfully request you reconsider the manuscript taking this into account. Specific responses are below:
>
> > However, in practice the situation in the test images could be much more complex. I wonder if the method would still work in these common and important situations.
>
> We completely agree that it is important to consider practical situations where the noise is complex. Most of our experiments correspond to controlled experiments because this allows for rigorous, quantitative evaluation of the behavior of the proposed method. However, we have also included results on a real-world application to transmission electron microscope (Section 5.4). These data are corrupted by spatially-variant shot noise, as well as other noise processes such as readout noise and dark current. Our results (see Figures 2, SM9, SM10)  show that GainTuning is very effective in denoising these real-world data, clearly outperforming alternative approaches. We will make sure that this is clear in the revised version of the manuscript.
>
> Inspired by this comment, we have also decided to include an additional controlled experiment where the noise behavior is qualitatively different to the noise model used for pretraining: we have applied GainTuning to a CNN pre-trained on additive Gaussian noise (which has fixed constant variance) to test data corrupted by Poisson noise (where the variance depends on the underlying pixel values and is hence spatially variant). GainTuning produces a substantial improvement in the denoising performance of the CNN (around 3.5 dB on Set12. See [table here](https://drive.google.com/file/d/1sKwPh6_rzpU2fMMjlhGVenpSL5mGboXe/view?usp=sharing) for more details).  These results are obtained using GainTuning based on the blind-spot loss on a UNet architecture [47] pre-trained on natural images with Gaussian noise of standard deviation between 0 and 55. We will include a more comprehensive version of this experiment in the revised version of the paper.
>
>
> > I suspect the original network needs to be big and powerful so that it may offer sufficient flexibility to the proposed approach. This may bring significant difficulty to network compression and quantization if one was to deploy such technology in real life.
>
> This is an interesting point. In our experiments we apply GainTuning to several standard network architectures from the literature (eg. [35, 47, 64, 65]), which are not particularly large. To be precise, we use DnCNN [64] and UNet [65] models with 668K and 233K parameters respectively, which are about 7-20x smaller than networks like MobileNet (~4.2M params. See Table 8 [here](https://arxiv.org/pdf/1704.04861.pdf) ) designed for practical applications. We will clarify this in the revised version of the paper.

---

> > ### Comment · Reviewer_aT3f · 2021-08-18
> > **One more question.**
> >
> > Thanks for the rebuttal. Could you please provide more intuition on why the proposed method works well for spatially varying noise? If this could be well justified (not just by experiment results), I think the authors addressed most of my concerns.

---

> > > ### Author Response · Authors · 2021-08-18
> > > **Explanation for noise generalization**
> > >
> > > This is a great question, we agree that providing a justification of why GainTuning is effective for spatially varying noise would be a good addition to the paper. This is very related to generalization across different noise levels. In fact, given the convolutional structure and the limited spatial field of view of state-of-the-art CNNs (eg. DnCNN [63] has a field of view of 41 x 41 pixels), a network that is able to denoise effectively at different unknown noise levels will also perform well for spatially varying noise (see for example Figure 12 in [Ref. [63] ](https://arxiv.org/pdf/1608.03981.pdf)). Understanding the performance of GainTuning for spatially varying noise, therefore, requires understanding how it adapts to different noise levels. Ref. [35] reports that CNNs tend to overfit the noise levels at which they have been trained, and provides an analytical explanation. They perform a first-order Taylor decomposition of the denoising CNN to show that overfitting occurs due to the presence of a nonzero additive term (called equivalent bias). [35] also shows that removing this term results in strong generalization across noise levels unseen during training. In Section 6 (lines 282-287), we demonstrate that GainTuning automatically implements this strategy. Optimizing the gain parameters reduces the equivalent bias of the denoiser, enabling generalization to new noise levels (see Section SM 7.3 and Figure SM 12). In fact, this fundamental change in the denoising function enables a CNN gain-tuned on one noisy image to generalize to other noisy images with similar noise levels as well (see lines 280-281 and Section SM 7.2). We will emphasize the implications of this analysis for spatially-varying noise in the revised version of the paper (namely that tuning the gain parameters results in a specific modification of the learned denoising function- reducing its equivalent bias- which renders it robust to varying noise levels, and hence ensures that it is able to handle spatially-varying noise).

---

> > > > ### Comment · Reviewer_aT3f · 2021-08-19
> > > > **more discussion**
> > > >
> > > > Your baseline CNN models are pre-trained on large datasets which I am not sure if it performs well with spatially varing noise if the model does not have knowledge on how the noise is going to be vary spatially. Then "GainTuning optimizes a single multiplicative scaling parameter (the “Gain”) of each channel in the convolutional layers of the CNN. How a single scaling parameter for each channel (I think this single parameter is used for the whole image during inference) help to improve the spatially varying denosing ability if the baseline may also have issues in it? Even if the strong baseline works fine in some images for the spatially varying noise, how does a single scaling parameter for each channel (which is used for the whole image during inference) help to improve the spatially varying denosing ability?

---

> > > > > ### Author Response · Authors · 2021-08-19
> > > > > **Detailed explanation for spatially variant noise generalization.**
> > > > >
> > > > > Thank you for the follow up questions. We will start by providing some background, and respond to the specific questions after.
> > > > >
> > > > > ### Convolutional structure and spatially varying noise
> > > > > +   As mentioned in our previous answer, CNNs trained for natural image denoising have a limited spatial receptive field - that is, they look at a region around a noisy pixel to compute the denoised value. For the DnCNN architecture this area is limited to 41x41. Furthermore, as can be seen by visualization of equivalent filters in Fig 4 of [Ref. [35]](https://arxiv.org/pdf/1906.05478.pdf) (and Fig 5 in our paper), the CNNs typically use an even smaller spatial region to perform denoising. As a reminder, these equivalent filters correspond to a first order Taylor expansion or linearization of the denoising function.
> > > > > +  CNNs implement highly non-linear functions, which depend heavily on the input. Since the network only looks at a small region in the input image to compute each denoised pixel, it ends up computing very different functions at different parts of the image. This is visualized through the equivalent filters in Fig 4 of Ref. [35] (and Fig 5 in our paper).
> > > > > + Because of the two previous points, denoising CNNs have the capability of dealing with spatially varying noise. For example, if a CNN is trained for all noise levels in [0, 55] and there is an input image with noise level 25 on the top and 15 on the bottom, the CNN will be able to denoise it effectively. An example for this is shown in Figure 12 of [Ref. [63] ](https://arxiv.org/pdf/1608.03981.pdf)).
> > > > >
> > > > > >  Your baseline CNN models are pre-trained on large datasets which I am not sure if it performs well with spatially varing noise if the model does not have knowledge on how the noise is going to be vary spatially.
> > > > >
> > > > > The pre-trained models we used from the literature are trained with a range of noise variances between 0 and 55. If the spatially varying noise has variance within this range, the network can handle it. However, if the noise is outside this range, the network may not be able to handle it (see examples below).
> > > > >
> > > > > ### Using GainTuning on spatially variant noise.
> > > > >
> > > > > >  how does a single scaling parameter for each channel (which is used for the whole image during inference) help to improve the spatially varying denosing ability
> > > > >
> > > > > **High-level answer.** As explained in the first part of the answer, CNNs are highly non-linear and can implement very different functions at different spatial regions of the image. We learn only a parameter for each channel, but there are multiple channels within a layer, and multiple layers are cascaded together (For DnCNN, we learn about 1,200 gain parameters). This provides flexibility to the denoising function, enabling it to implement different denoising functions in different parts of the image.
> > > > >
> > > > > **Detailed answer with examples.**
> > > > >
> > > > > As shown in Figure 12 of [Ref. [63] ](https://arxiv.org/pdf/1608.03981.pdf) if the image has spatially varying noise and all noise levels are within the training range of the CNN, the CNN is able to denoise it effectively. However, if there is spatially varying noise and some of the noise levels are outside the training range, the pre-trained CNN fails catastrophically in parts of the image with out-of-distribution noise. Consider [this image where the left part is corrupted with a noise level of 30 and the right part of the image is corrupted with noise level of 90](https://drive.google.com/file/d/1vrZ94qOGydNy7ZJhNN4PPYyc-0duUW92/view?usp=sharing). A DnCNN pre-trained on noise levels 0-55 fails on the right part of the image. However, after performing GainTuning (on the entire image), the CNN generalizes to this image.
> > > > >
> > > > > +   **Why did the pre-trained CNN fail in this example?** Ref. [35] reports that CNNs tend to overfit the noise levels at which they have been trained, and provides an analytical explanation. They perform a first-order Taylor decomposition of the denoising CNN to show that overfitting occurs due to the presence of a nonzero additive term (called equivalent bias). This observation is true for each spatial region: For regions where the noise is outside the training range the CNN has a large equivalent bias. For regions where the noise is within the training range the CNN has smaller equivalent bias.
> > > > >
> > > > >  + **How does GainTuning generalize in this situation?** Ref.[35] showed that removing bias terms from CNN enables it to generalize to out-of-distribution noise. In Section 6 (lines 282-287), we demonstrate that GainTuning automatically implements this strategy. Optimizing the gain parameters reduces the equivalent bias of the denoiser, enabling generalization to new noise levels (see Section SM 7.3 and Figure SM 12).
> > > > >
> > > > >  +  **Why does reducing bias help with spatially variant out-of-distribution noise?** As shown in Figure SM 12 (and in Ref.[35]) equivalent biases of CNNs are small at noise levels where they denoise effectively. When GainTuning is performed at a given out-of-distribution noise level, it does not just reduce the equivalent bias at that noise level, but it also reduces equivalent bias at other noise levels. Fig SM 12 shows an example of performing GainTuning on a noisy image at $\sigma=80$ on a pre-trained denoised train for $\sigma \in [0, 55]$. The equivalent bias at $\sigma=80$ is reduced, but the equivalent bias at all the other noise levels also goes down. Now, when this network encounters an input region with a different noise level, it will generalize well to that noise level as well.  We show this with [an example of an image where 4 different parts have 4 out-of-distribution noise levels](https://drive.google.com/file/d/1GJ054Qcet943lvizghDHNrSsvwYBnnJB/view?usp=sharing). Top left: 60, Top right: 70. Bottom left: 80. Bottom right: 90. The pre-trained DnCNN is trained for 0-55, do not denoise well. GainTuning denoises well and achieves around 4dB improvement in performance.

---

> > > > > > ### Comment · Reviewer_aT3f · 2021-08-20
> > > > > > **reasonable but could be better**
> > > > > >
> > > > > > I think the authors' explanation is reasonable and I tend to believe that this paper may be published. However, one thing that still bothers me is the proposed method only optimized one parameter for each channel globally, then the results seems to be dependent on the receptive field of the network. It is not clear a bigger receptive field or a small one will make the network do better or worse. In sum, I think the idea is interesting but the decision of optimizing one para for each channel seems a bit adhoc and deserves more thorough considerations. Overall, the paper is borderline.

---

> > > > > > > ### Author Response · Authors · 2021-08-20
> > > > > > > **Additional clarifications**
> > > > > > >
> > > > > > >
> > > > > > > Thank you for the additional comments, which raise some points that we believe are worth clarifying.
> > > > > > >
> > > > > > > > method only optimized one parameter for each channel globally, then the results seems to be dependent on the receptive field of the network.
> > > > > > >
> > > > > > > We mentioned the receptive fields of denoising CNNs in our previous answer in order to explain how GainTuning is able to handle spatially varying noise. We would like to clarify that the limited receptive field of denoising CNNs does not have anything to do with GainTuning specifically. The receptive field necessary to perform good denoising will depend on the underlying signal. Previous work has shown that for natural images a small receptive field is sufficient (see for example Table 3.1 in Ref. [36]). We evaluate GainTuning on several network architectures published in the literature ( eg. [63, 35, 47, 27]) on multiple test conditions (eg. out-of-distribution noise, or signal) and show that it performs well for all of them. This indicates that GainTuning does not only work for one network architecture, but is in fact more broadly applicable.
> > > > > > >
> > > > > > >
> > > > > > >
> > > > > > > > I think the idea is interesting but the decision of optimizing one para for each channel seems a bit adhoc and deserves more thorough considerations.
> > > > > > >
> > > > > > > We agree with the reviewer that it would be good to include a more thorough discussion on the choice to fine-tune gains.  We will also add the experiments below, showing that other fine-tuning choices yield inferior results.
> > > > > > >
> > > > > > >
> > > > > > > **Motivation**:  Adjusting gain parameters provides enough flexibility to modify the denoising function at test time, while constraining the complexity of this modification to avoid overfitting. Use of this specific parameter subset is motivated by the following observations:
> > > > > > > +  The first order moments of the feature maps of each channel in intermediate layers of classification networks have been shown to capture style-related properties [ 17, [Ding et. al. 2020](https://arxiv.org/pdf/2004.07728.pdf)]. We can interpret this to mean that they can be used to adjust image-style priors. Adjusting gains directly modifies these moments.
> > > > > > > + As pointed out in Section 2, adjustment of channel parameters like gain has been shown to help in generalization in multiple deep learning problems [41, 11, 7, 17, 20, 25, 39, 48] (see also the related work pointed out by Reviewer sYYW). In most of these examples, the channel parameters were adjusted during training. In GainTuning they are optimized at test time.
> > > > > > > + Gain adjustment in a CNN is a natural generalization of the concept of “whitening” in a linear system, in which one rescales the data after projecting onto the eigenvectors. This sort of rescaling or gain adjustment is popular in many traditional denoising algorithms (eg. Wiener filter performs gain adjustment in Fourier domain, many wavelet based methods change the gain of coefficients after the forward transformation).
> > > > > > > + CNN gains are usually adapted during training (ie., batch norm), to keep network responses in a reasonable range. But biological systems adjust gains during operation, at multiple time scales, to adapt to context and improve representational efficiency  (e.g., [5]).
> > > > > > >
> > > > > > >
> > > > > > > **Experimental motivation**. In order to further justify the choice of fine-tuning gain parameters, we compare against fine-tuning only the last few layers of the network (this is a popular choice for fine-tuning deep learning models in image recognition). In particular, we selectively fine-tuned only the last n (n=1, 2, 3, 4, and 10) layers of a DnCNN (with 20 layers) for test images (1) that are in-distribution, (2) corrupted with out-of-distribution noise, and (3) contain image content different from training data. We summarize the results in a [table here](https://drive.google.com/file/d/1l60dT126-EGUc53ccBRn4IeTp8wAL5fS/view?usp=sharing) (see [boxplot](https://drive.google.com/file/d/1pBjndhZy2gM2_NAdIHm71POtQL8Jqyt1/view?usp=sharing) for the distribution of improvements for out-of-distribution noise experiment). Note that the entries in the table are improvements in PSNR which is the difference between performance after GainTuning and the pre-trained network (large positive values are better). Fine-tuning only the gains outperforms all the other alternatives.

---

### Decision · Program_Chairs · 2021-09-28

**Decision:**

Accept (Poster)

**Comment:**

  To solve the image denoising problem, this paper proposed to tune a single gain parameter for each channel on a single test image. The tuning is done via minimizing an unsupervised loss. Compared to non-fine-tuning methods, tuning can provide better adaptation to out-of-distribution images. Compared to other tuning-methods, this approach is designed to avoid the overfitting issue of fine-tuning all hyperparameters. The idea is simple and natural, and this work has presented many experiments to show that it outperforms existing methods. After rebuttal, all reviewers think that the rebuttal has addressed their most concerns, and agree with the acceptance (though some are on the borderline). Thus I recommend acceptance.

  Note that one reviewer requested to add experimental comparison with conditioned models such as DVDnet. The goal of the reviewer is to understand the difference with the works that study image restoration by providing info about the degradation to the restoration neural-net. It will be good to see which method is better.

**Consistency Experiment:**

NeurIPS has a long history of experimentation. In 2014, NeurIPS ran an experiment in which 10% of submissions were reviewed by two independent committees to quantify the randomness in the review process. This year, we repeated a variant of this experiment to see how the quality of the review process has changed over time.  This paper was part of the experiment and was therefore assigned to two committees (consisting of reviewers, an Area Chair, and a Senior Area Chair) that reached independent decisions.  If both committees made the same recommendation, this recommendation was followed. If a single committee recommended acceptance, the paper was accepted (with the exception of a few cases in which the other committee identified what we considered a fatal flaw, e.g., an error in a key result).

This copy’s committee reached the following decision: **Accept (Poster)**

The other committee assigned to the paper recommended **Reject**.  You can find the other set of reviews, along with any follow up discussion with the authors here:
https://openreview.net/forum?id=vmJs9dyUeWQYe